# Limited effect of the confluence angle and tributary gradient on Alpine confluence morphodynamics under intense sediment loads

**Authors:**

**Théo St. Pierre Ostrander**[(1)] – Corresponding Author

theo.st-pierre-ostrander@uibk.ac.at

**Thomé Kraus**[(2)]

Thome.Kraus@student.uibk.ac.at

**Bruno Mazzorana**[(3)]

bruno.mazzorana@uach.cl

**Johannes Holzner**[(4)]

Johannes.Holzner@provinz.bz.it

**Andrea Andreoli**[(5)]

Andrea.Andreoli@unibz.it

**Francesco Comiti**[(6)]

Francesco.Comiti@unibz.it

**Bernhard Gems**[(7)]

bernhard.gems@uibk.ac.at

[(1, 2, 7)] University of Innsbruck, Unit of Hydraulic Engineering, Technikerstraße 13a, 6020, Innsbruck, Austria

[(3)] Universidad Austral de Chile, Faculty of Sciences, Instituto de Ciencias de la Tierra, Edificio Emilio Pugín, Avenida Eduardo Morales Miranda, Campus Isla Teja, Valdivia, Chile

[(4, 5, 6)] Free University of Bozen-Bolzano, Faculty of Agricultural, Environmental and Food Sciences, Universitätsplatz 5 - piazza Università, 5, 39100, Bozen-Bolzano, Italy

**Abstract**
Confluences are dynamic morphological nodes in all river networks. In mountain regions, they are
influenced by hydraulic and sedimentary processes occurring in steep channels during extreme events in
small watersheds. Sediment transport in the tributary channel and aggradation in the confluence can be
massive, potentially causing overbank flooding and sedimentation into adjacent settlement areas. Previous
works dealing with confluences have been mainly focused on lowland regions, or if focused on mountain
areas, the sediment concentrations and channel gradients were largely under-representative of mountain
river conditions. The presented work contributes to filling this research gap with 45 experiments using a
large-scale physical model. Geometric model parameters, applied grain size distribution, and the
considered discharges represent the conditions at 135 confluences in South Tyrol (Italy) and Tyrol (Austria).
The experimental program allowed for a comprehensive analysis of the effects of (i) the confluence angle,
(ii) the tributary gradient, (iii) the channel discharges, and (iv) the tributary sediment concentration. Results
indicate, in contrast to most research dealing with confluences, that in the presence of intense tributary
sediment supply and a small tributary to main channel discharge ratio (0.1), the confluence angle does not
have a decisive effect on confluence morphology. Adjustments to the tributary channel gradient yielded the
same results. A reoccurring range of depositional geomorphic units was observed where a deposition cone
transitioned to a bank-attached bar. The confluence morphology and tributary channel gradient rapidly
adjusted, tending towards an equilibrium state to accommodate both water discharges and the sediment
load from the tributary. Statistical analyses demonstrated that confluence morphology was controlled by the
combined channel discharge and the depositional or erosional extents by the sediment concentration.
Applying the conclusions drawn from lowland confluence dynamics could misrepresent depositional and
erosional patterns and the related flood hazard at mountain river confluences.
**Keywords:** Confluence Morphology; Fluvial Hazard; Steep Tributary; Bedload; Physical Scale
Model; Mountain Rivers

## 1   Introduction

River confluences are important features of all river systems and are sites of significant hydraulic and morphological change (Benda et al., 2004). They are characterized by converging flow paths that produce complex 3-dimensional hydraulics that influence the local morphology, and fluvial dynamics (Best, 1987; 1988; Rhoads & Kenworthy, 1995; Benda et al., 2004; Boyer et al., 2006; Ferguson & Hoey, 2008; Guillén-Ludeña et al., 2015; Guillén-Ludeña et al., 2017). In developed areas, confluences form critical junctions as the hydraulic geometries and sediment loads from each channel must be accommodated to avoid overbank flooding and sedimentation (Gems et al., 2014; Liu et al., 2015; Kammerlander et al., 2016; Sturm et al., 2018). The importance of these junctions has garnered much research interest, which has illuminated many characteristics of the hydro-morphodynamic interactions, and the major controls on the flow structure occurring at lowland river confluences (Mosley, 1976; Best, 1987; 1988; Biron et al., 1993; Rhoads & Kenworthy, 1995; Bradbrook et al., 1998; De Serres et al., 1999; Benda et al., 2004; Boyer et al., 2006; Wang et al., 2007; Liu et al., 2015). Best (1987; 1988) built upon the seminal work of Mosley (1976) in his identification of hydraulic and morphologic zones occurring at confluences. The typically occurring hydraulic zones are: flow separation, flow stagnation, flow deflection, maximum velocity, shear layers, and the recovery zone. These zones influence sediment transport pathways through the confluence and the resulting morphological elements of confluences: avalanche faces at the mouth of each confluent channel, a deep central scour hole, and a bar in the separation zone. Best (1988) concluded that the controlling variables as to the location, orientation, and size of these morphologic zones are the confluence angle and the discharge ratio $Q_r = Q_t/Q_m$ which is the ratio of the tributary ($Q_t$) and the main channel ($Q_m$) discharges. For lowland confluences increasing the discharge ratio or the confluence angle leads to a greater mutual deflection of flows and a bigger separation zone, which is the largest sink for tributary-transported sediment (Best, 1987, 1988). Flow deflection influences the shear layers generated between the two convergent flows, along which powerful vortices are generated which are responsible for increased bed shear stresses in the junction (Mosley, 1976; Best, 1987; Penna et al., 2018; De Serres et al., 1999). Contrarily, decreasing the confluence angle results in a greater mixing of flows, a smaller separation zone, and declined levels of turbulence in the confluence (Best, 1988; Penna et al., 2018). However, mountain channels are steeper than lowland channels with higher velocities and supercritical flows that amplify event intensity (Rudolf-

Miklau et al., 2013) and can result in rapid channel adjustments (Wohl, 2010). This is apparent when
comparing, for example, the Froude numbers from Best (1988) (0.1-1) and Biron et al. (1996) (0.1-0.24),
and the tributary velocities (0.45 m s$^{-1}$-0.57 m s$^{-1}$) from Roy and Bergeron (1990) with the Froude numbers
and velocities from the presented work (Table 1) and steep channels in the study region (e.g., Hübl et al.,

82 2005).


**Table 1** Experimental discharges for the main ($Q_m$) and tributary ($Q_t$) channels with corresponding hydraulic
attributes showing flow depth ($h$), Froude ($Fr$), and velocity ($v$) upstream ($u$) and downstream ($d$) of the
confluence and in the tributary channel ($t$), for all confluence angles (CA) and tributary gradients (trib.),
values are based on undisturbed, initial conditions in the channel.

| | $Q_m$ | $Q_t$ | $Q_{tot}$ | $h_u$ | $h_t$ | $h_d$ | $Fr_u$ | $Fr_t$ | $Fr_d$ | $v_u$ | $v_t$ | $v_d$ |
|---|---|---|---|---|---|---|---|---|---|---|---|---|
| | [l s$^{-1}$] | [l s$^{-1}$] | [l s$^{-1}$] | [m] | [m] | [m] | [-] | [-] | [-] | [m s$^{-1}$] | [m s$^{-1}$] | [m s$^{-1}$] |
| CA 90° Trib. 10% [EXP 1-15] | 15 | 1.5 | 16.5 | 0.04 | 0.01 | 0.03 | 0.58 | 2.04 | 0.77 | 0.35 | 0.68 | 0.44 |
| | 45 | 4.5 | 49.5 | 0.08 | 0.02 | 0.06 | 0.53 | 2.39 | 0.98 | 0.47 | 1.08 | 0.75 |
| | 75 | 7.5 | 82.5 | 0.11 | 0.03 | 0.08 | 0.59 | 2.79 | 1.00 | 0.61 | 1.43 | 0.89 |
| | 105 | 10.5 | 115.5 | 0.14 | 0.04 | 0.10 | 0.62 | 2.63 | 1.01 | 0.73 | 1.52 | 1.01 |
| | 135 | 13.5 | 148.5 | 0.17 | 0.04 | 0.12 | 0.66 | 2.87 | 1.06 | 0.84 | 1.76 | 1.16 |
| CA 90° Trib. 5% [EXP 16-30] | 15 | 1.5 | 16.5 | 0.05 | 0.01 | 0.04 | 0.46 | 1.55 | 0.69 | 0.31 | 0.57 | 0.42 |
| | 45 | 4.5 | 49.5 | 0.09 | 0.03 | 0.07 | 0.50 | 1.79 | 0.80 | 0.47 | 0.90 | 0.71 |
| | 75 | 7.5 | 82.5 | 0.12 | 0.04 | 0.09 | 0.51 | 1.84 | 1.02 | 0.56 | 1.08 | 0.93 |
| | 105 | 10.5 | 115.5 | 0.15 | 0.04 | 0.11 | 0.52 | 1.82 | 1.04 | 0.63 | 1.19 | 1.04 |
| | 135 | 13.5 | 148.5 | 0.18 | 0.05 | 0.13 | 0.52 | 1.90 | 0.97 | 0.69 | 1.34 | 1.08 |
| CA 45° Trib. 10% [EXP 31-45] | 15 | 1.5 | 16.5 | 0.04 | 0.01 | 0.04 | 0.56 | 1.79 | 0.69 | 0.35 | 0.60 | 0.42 |
| | 45 | 4.5 | 49.5 | 0.08 | 0.02 | 0.07 | 0.68 | 2.24 | 0.71 | 0.58 | 1.04 | 0.70 |
| | 75 | 7.5 | 82.5 | 0.11 | 0.03 | 0.09 | 0.61 | 2.54 | 0.96 | 0.64 | 1.34 | 0.89 |
| | 105 | 10.5 | 115.5 | 0.14 | 0.04 | 0.11 | 0.60 | 2.52 | 0.90 | 0.70 | 1.48 | 0.94 |
| | 135 | 13.5 | 148.5 | 0.16 | 0.04 | 0.13 | 0.61 | 2.77 | 0.95 | 0.77 | 1.72 | 1.07 |


Confluences in mountain regions have not received the same attention as those in lowland areas, which is
surprising given the hazard potential associated with large volumes of coarse sediment entering these
critical junctions (Aulitzky, 1989). Differentiation between mountain and lowland confluences can be
described by (i) supercritical or transitioning flow conditions in the tributary channel, (ii) bed surface armoring
due to the size heterogeneity of the tributary sediment load or non-erodible conditions in the tributary
channel as a result of hazard protection measures, (iii) high sediment concentrations during flooding events
and (iv) highly variable discharges and sediment transport rates (Aulitzky, 1980; 1989; Meunier, 1991; Roca
et al., 2009; Guillén-Ludeña et al., 2017). Topographic confinement can amplify confluence effects, whereas
in lowland regions with wide valley floors and broad terraces, deposition cones or fans can be isolated from
the main channel (Benda et al., 2004). A sudden introduction of sediment from steep tributaries can trigger
numerous types of morphological changes (Benda et al., 2004), as tributaries of confined channel
confluences can be particularly impactful (Rice, 1998).
Detailed records of flash flooding associated with intense sediment transport in Tyrol (Austria) show that
these events are a persistent hazard (Embleton-Hamann, 1997; Rom et al., 2023). In the Alps, hazardous
events can impact high-population-density valleys. Increased or shifting flooding patterns (Blöschl et al.,
2017; Löschner et al., 2017; Blöschl et al., 2020; Hanus et al., 2021) and enhanced sediment availability
(Knight & Harrison, 2009; Stoffel et al., 2012; Gems et al., 2020) as a consequence of climate change (Keiler
et al., 2010) not only threatens new infrastructure but challenges previously installed mitigation measures.
Ancey (2020a) discusses the complications, and assumptions associated with the multitude of approaches
used to predict bedload transport and the resulting bedforms, and how rivers are systems punctuated by
intense moments of bedload transport resulting in rapid changes in bed morphology over short time intervals
(Ancey, 2020b). Relevant hazard events are typically triggered by localized short-duration-high intensity
convective storms occurring in small watersheds, which do not significantly affect main channel discharge
and bedload transport (Gems et al., 2014; Hübl & Moser, 2006; Prenner et al., 2019; Stoffel, 2010). The
narrow, steep tributary provides the sediment load to the main channel, which supplies the dominant flow
discharge (Miller, 1958; Guillén-Ludeña et al., 2017).
Most of the work that has been done on mountain river confluences has been focused on conditions that
do not typically generate hazardous events, mainly under-representations of gradients and sediment
concentrations (Roca et al., 2009; Leite Ribeiro et al., 2012a; Leite Ribeiro et al., 2012b; Guillén-Ludeña et
al., 2015; Guillén-Ludeña et al., 2017). Complicating the conclusions drawn regarding confluence
morphodynamics, St. Pierre Ostrander et al. (2023) established, from a set of 15 experiments, that
confluences of mountain rivers are influenced by factors other than the confluence angle and the discharge
ratio. They held the confluence angle and discharge ratio constant, only adjusting discharges and tributary
sediment concentration. They observed a range of morphologies with specific geomorphic units: a
deposition cone, a transitional morphology, a bank-attached bar, and a scour hole. They used unit stream
power to predict and associate confluence zone morphology with hydraulic conditions. However, they were
limited in their conclusions and recommended further experiments considering additional geometries as
their experimental program was not sufficiently comprehensive, restricting the reach of their findings. The
channel geometry was unchanged throughout the experimental program, and morphological assessment
lacked statistical evaluation and grain size analysis. This paper builds upon these experimental results with
an additional 30 experiments considering geometric modifications. In addition to investigating the effects of
the channel discharge and sediment concentration, adjustments to the confluence angle and the tributary
gradient provide a more comprehensive data analysis of fluvial hazard processes and the resulting
morphologies of mountain river confluences. Evaluating morphological patterns and extents was done
qualitatively with DEMs of Difference (DoD) created from laser scans, quantitatively from the extents of
geomorphic units, depositional and erosional values, and volumetric grain samples, and statistically.
Statistical analyses determined which of the introduced controlling factors significantly impacted the
response variables that define the morphodynamic development of mountain river confluences. Results
from the 45 experiments tested the following hypotheses:
1. Adjustments to the confluence angle and the tributary gradient do not significantly impact

confluence morphology and the development of specific geomorphic units (hypothesis 1).

2. Of the introduced controlling factors, the sediment concentration and channel discharge exert the

most control over depositional and erosional patterns (hypothesis 2).

The formulation of the two hypotheses was based on the results of St. Pierre Ostrander et al. (2023) where
it was established that in addition to the confluence angle and discharge ratio, there were additional factors
influencing the morphological development of the confluence, and from a review of literature dealing with
rivers in response to intense hydrological events. Specifically, a channel will adjust its geometric
characteristics and gradient in a way that maximizes sediment transport capacity (Lane, 1955; White et al.,

1982).

## 2   Model and Methods

### 2.1   Experimental program

The physical scale model (Fig. 1) was constructed to represent a typical confluence in the regions of South Tyrol (Italy) and Tyrol (Austria). The experimental setup served as a generic configuration to reproduce the main hydrodynamic and sedimentary processes occurring at mountain river confluences while gaining insights into the dominant control variables. Experimental modeling uses and builds upon the configuration, calibration, and experiments (1-15) carried out by St. Pierre Ostrander et al. (2023), but considers an additional case for the tributary gradient as well as for the confluence angle. Model dimensions, discharges, and the grain size distribution of the quartz sand input material and the main channel bed were based on an analysis of 135 confluences and 65 volume (subsurface) and line (surface) sediment samples in the study region (St. Pierre Ostrander et al., 2023). The sediment mix was scaled by a factor of 30 to transfer natural grain size dimensions to model conditions. The main channel had a mobile bed, allowing for 0.2 m of erosion while the tributary channel had a fixed bed. Tributary bed roughness was created using an adhesive to apply a layer of quartz sand to the bed. Channel roughness was established through hydraulic manuals (Chow, 1959) and previous calibration work (St. Pierre Ostrander et al., 2023). Quartz sand is widely used in flume experiments dealing with gravel bed rivers (e.g., Williams, 1970; Gems et al., 2014), as the grain density ($\rho_s$=2650 kg m$^{-3}$) supports Froude model similitude (Young & Warburton, 1996). A grain size distribution curve and the gradation coefficient ($\sigma$) of both the mobile bed and the input material are included in Fig. 1. The physical model was adjustable, except for the width of the tributary (0.2 m) and the lengths of the channels (5.0 m and 9.0 m for the tributary and main channel, respectively). Discharge to each channel was supplied by separate pumps controlled by electronic flow measurement devices. The discharge ratio was fixed at 0.1 for all experiments. The tributary sediment discharge was always proportional to the clear water discharge; an increase in tributary discharge meant an increase in both clear water and sediment discharges. The main channel flow was exclusively clear water and fully rough turbulent to replicate typical events that produce massive aggradation at mountain river confluences (Hübl & Moser, 2006; Stoffel, 2010; Gems et al., 2014; Prenner et al., 2019). Scaling was done according to Froude similarity; transferring model dimensions to nature allows a scale factor range of 20-40. The scale is

determined by the width of the tributary at the confluence relative to the width of the tributary in the physical
model and is referred to as the specific scale (St. Pierre Ostrander et al., 2023).

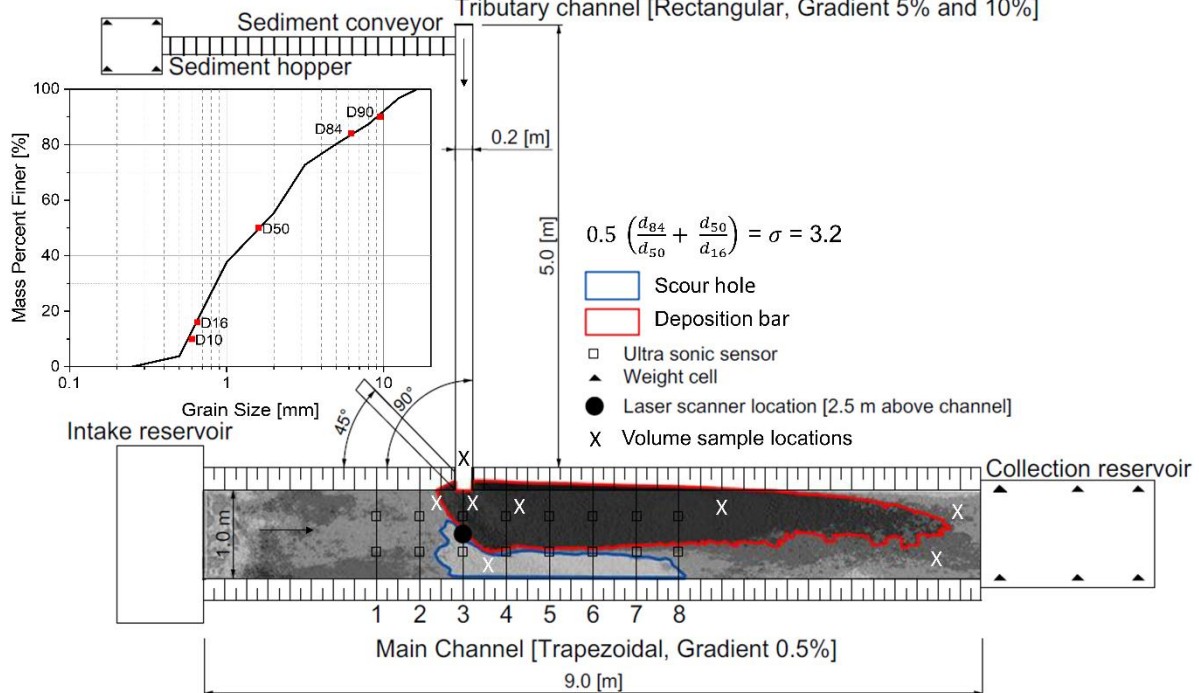


**Figure 1** Overview of the physical model showing the location of measurement devices, volume sample
locations, the gradation coefficient ($\sigma$), the grain size distribution of the sediment supplied to the tributary
channel and the mobile bed in the main channel, and an example of the scour hole and the deposition bar.

Experiments (Table 2) allowed for the same 5 steady-state discharge combinations to be tested with
different tributary gradients, confluence angles, and sediment concentrations, which were based on the bulk
density of the input material. The 5 discharges correspond to flooding conditions in the study region,
including an extreme event. Steady-state discharges were used so a specific discharge could be linked with
a geomorphic unit, to limit uncertainty in associating morphologies with the introduced controlling factors,
which is consistent with other researchers dealing with steep channel confluences, (Roca et al., 2009; Leite
Ribeiro et al., 2012), and to make the morphological development comparable to research dealing with
lowland confluences, which largely assume steady-state conditions (e.g., Mosley, 1976; Best, 1988). The
morphological development of the confluence zone for each geometric setup was evaluated by creating
DEMs of Difference (DoD) (*ESRI ArcGIS Desktop, Release 10.8.2*) from laser scans (*Faro Focus 3D,*
*Trimble X7*) taken before and after an experiment. Each laser scan contained 125 million points with a point
density of 0.004 m at a distance of 10 m. The average error between the position of the scanner and the
targets used for referencing the scans was less than 0.004 m. The initial bathymetry was the reference,
which was established by running a low discharge of 15 l s$^{-1}$ in the main channel for 5 hours to create a
more natural river bed, while the post-run bathymetry was the comparison (St. Pierre Ostrander et al., 2023).
Morphological evaluation was done by assessing specific zones and overall changes occurring in the
channel. The deposition bar and scour hole were delineated by deposition or erosion above or below 0.01
m (Fig. 1). Main channel deposition and erosion areas and volumes reflect morphological change occurring
throughout the entire channel above or below the initial bathymetry.
Based on incident reports supplied by the Austrian Service for Torrent and Avalanche Control and event
documentation (e.g. Hübl et al., 2012), the scaled (30), according to Froude similarity, experiment duration
was 20 minutes and started when sediment entered the tributary channel. The only alterations between the
experimental groups were changing the tributary gradient and the confluence angle. Experiments 1-15 had
a 10% tributary gradient, a 90° confluence angle, and a main channel gradient of 0.5%. Experiments 16-30
had the same geometric configuration except with a 5% tributary gradient. Experiments 31-45 had a 10%
tributary gradient and a 45° confluence angle; the main channel gradient remained unchanged. The
respective dimensions were chosen as they are the most representative of the study region (St. Pierre
Ostrander et al., 2023). DEMs of Difference were created from the DoDs of experiments with identical input
conditions, i.e., discharge and sediment supply rate, allowing for a visual assessment of morphological
differences based on geometric changes alone. For example, experiments 1 and 16 had equal discharges
and sediment concentrations; the only change was the tributary gradient, and experiments 1 and 31 had
the same discharges, sediment concentrations, and gradients, but the confluence angle was changed. The
10% gradient tributary with a 90° confluence angle was used as the reference as both geometric
configurations are comparable, and changes from the gradient and confluence angle could be accurately
assessed.
**Table 2** Experiment target and actual discharges and sediment concentration, and tributary sediment supply
rate, $Q$ denotes discharge while $m$ or $t$ subscripts refer to the main channel and the tributary channel,
respectively. The main channel gradient was 0.5% for all experiments. Experiment 30 could not be
completed as the deposition in the tributary caused overtopping of the channel.

| | EXP | $Q_m$ Target | $Q_m$ Actual | $Q_t$ Target | $Q_t$ Actual | Sed. conc. Target | Sed. conc. Actual | Sed. supply rate |
|---|---|---|---|---|---|---|---|---|
| | [-] | [l s$^{-1}$] | [l s$^{-1}$] | [l s$^{-1}$] | [l s$^{-1}$] | [%] | [%] | [kg min$^{-1}$] |
| **10% Tributary Gradient 90° Confluence Angle** | 1 | 15.0 | 15.3 | 1.5 | 1.5 | 5.0 | * | 7.6 |
| | 2 | 45.0 | 45.6 | 4.5 | 4.3 | 5.0 | * | 22.9 |
| | 3 | 75.0 | 75.5 | 7.5 | 7.4 | 5.0 | 5.7 | 43.5 |
| | 4 | 105.0 | 104.5 | 10.5 | 10.6 | 5.0 | 4.9 | 53.4 |
| | 5 | 135.0 | 135.4 | 13.5 | 13.4 | 5.0 | 5.2 | 68.7 |
| | 6 | 15.0 | 15.1 | 1.5 | 1.5 | 7.5 | 7.6 | 11.4 |
| | 7 | 45.0 | 46.1 | 4.5 | 4.4 | 7.5 | 7.5 | 34.3 |
| | 8 | 75.0 | 75.3 | 7.5 | 7.5 | 7.5 | 7.3 | 57.2 |
| | 9 | 105.0 | 105.1 | 10.5 | 10.5 | 7.5 | 7.6 | 80.1 |
| | 10 | 135.0 | 134.7 | 13.5 | 13.4 | 7.5 | 7.5 | 103.0 |
| | 11 | 15.0 | 14.8 | 1.5 | 1.5 | 10.0 | * | 15.3 |
| | 12 | 45.0 | 44.9 | 4.5 | 4.6 | 10.0 | 10.1 | 45.8 |
| | 13 | 75.0 | 76.1 | 7.5 | 7.6 | 10.0 | 10.3 | 76.3 |
| | 14 | 105.0 | 105.7 | 10.5 | 10.4 | 10.0 | 10.4 | 106.8 |
| | 15 | 135.0 | 135.4 | 13.5 | 13.6 | 10.0 | * | 137.3 |
| **5% Tributary Gradient 90° Confluence Angle** | 16 | 15.0 | 15.9 | 1.5 | 1.4 | 5.0 | * | 7.6 |
| | 17 | 45.0 | 46.0 | 4.5 | 4.5 | 5.0 | 5.1 | 22.9 |
| | 18 | 75.0 | 75.9 | 7.5 | 7.6 | 5.0 | 5.0 | 43.5 |
| | 19 | 105.0 | 104.4 | 10.5 | 10.4 | 5.0 | 5.1 | 53.4 |
| | 20 | 135.0 | 134.7 | 13.5 | 13.5 | 5.0 | 5.2 | 68.7 |
| | 21 | 15.0 | 15.5 | 1.5 | 1.4 | 7.5 | * | 11.4 |
| | 22 | 45.0 | 46.7 | 4.5 | 4.3 | 7.5 | 7.8 | 34.3 |
| | 23 | 75.0 | 74.9 | 7.5 | 7.5 | 7.5 | 7.5 | 57.2 |
| | 24 | 105.0 | 105.5 | 10.5 | 10.4 | 7.5 | 7.5 | 80.1 |
| | 25 | 135.0 | 134.6 | 13.5 | 13.4 | 7.5 | 7.9 | 103.0 |
| | 26 | 15.0 | 15.1 | 1.5 | 1.6 | 10.0 | 9.6 | 15.3 |
| | 27 | 45.0 | 43.5 | 4.5 | 4.4 | 10.0 | 10.2 | 45.8 |
| | 28 | 75.0 | 75.0 | 7.5 | 7.6 | 10.0 | 10.1 | 76.3 |
| | 29 | 105.0 | 105.9 | 10.5 | 10.5 | 10.0 | 10.1 | 106.8 |
| | 30 | 135.0 | - | 13.5 | - | - | - | - |
| **10% Tributary Gradient 45° Confluence Angle** | 31 | 15.0 | 14.6 | 1.5 | 1.6 | 5.0 | * | 7.6 |
| | 32 | 45.0 | 45.0 | 4.5 | 4.3 | 5.0 | 5.2 | 22.9 |
| | 33 | 75.0 | 75.8 | 7.5 | 7.7 | 5.0 | 4.9 | 43.5 |
| | 34 | 105.0 | 105.1 | 10.5 | 10.5 | 5.0 | 5.0 | 53.4 |
| | 35 | 135.0 | 134.9 | 13.5 | 13.5 | 5.0 | 5.0 | 68.7 |
| | 36 | 15.0 | 15.0 | 1.5 | 1.5 | 7.5 | * | 11.4 |
| | 37 | 45.0 | 45.6 | 4.5 | 4.5 | 7.5 | 7.6 | 34.3 |
| | 38 | 75.0 | 75.2 | 7.5 | 7.5 | 7.5 | 7.7 | 57.2 |
| | 39 | 105.0 | 106.1 | 10.5 | 10.5 | 7.5 | 7.6 | 80.1 |
| | 40 | 135.0 | 135.6 | 13.5 | 13.4 | 7.5 | 8.0 | 103.0 |
| | 41 | 15.0 | 14.8 | 1.5 | 1.4 | 10.0 | 10.4 | 15.3 |
| | 42 | 45.0 | 44.9 | 4.5 | 4.4 | 10.0 | 10.1 | 45.8 |
| | 43 | 75.0 | 75.5 | 7.5 | 7.6 | 10.0 | 9.9 | 76.3 |
| | 44 | 105.0 | 105.8 | 10.5 | 10.4 | 10.0 | 9.3 | 106.8 |
| | 45 | 135.0 | 135.0 | 13.5 | 13.5 | 10.0 | * | 137.3 |

*indicates that the sediment was delivered manually or with manual assistance as the dosing machine could not dose
very low or high rates of sediment into the tributary channel

## 2.2 Statistical analysis

A statistical analysis of the various introduced controlling factors and their effects on the response variables
(Table 3) was done using the software package *OriginPro (v.2023, OriginLab Corp.)* (Stevenson, 2011;
Baranovskiy, 2019). The chosen response variables (Table 3), captured either depositional or erosional
features and allowed for a nuanced investigation into the subtle morphological variations that were not able
to be qualitatively assessed. The combined discharge was used as a factor since the morphological
development of the confluence occurred downstream of the tributary. The confidence interval for all tests
was 95%. A significant result occurred when the p-value, calculated from the test statistic of the applied
test, was less than 0.05. A p-value less than 0.05 allowed for rejecting the null hypothesis, which was the
factor that did not significantly impact the response variable. If rejected, further pairwise post hoc tests were
conducted to determine the decisive factors influencing confluence morphology.
**Table 3** Controlling factors and response variables that control and define confluence morphology.

| Factor | Unit | Response Variable | Unit |
|---|---|---|---|
| Sediment concentration (5, 7.5, 10) | % | Main channel deposition area and volume | $m^2$, $m^3$ |
| Combined discharge (16.5, 49.5, 82.5, 115.5, 148.5) | $l\,s^{-1}$ | Main channel erosion area and volume | $m^2$, $m^3$ |
| Confluence angle (90, 45) | ° | Deposition bar area | $m^2$ |
| Tributary gradient (10, 5) | % | Deposition bar length | m |
| | | Deposition bar width | m |
| | | Scour area | $m^2$ |
| | | Scour length | m |
| | | Scour width | m |
| | | Maximum depths scour and deposition | m |

The sequence of operations in Fig. 2 shows the chosen tests, which allowed for planned comparisons
(Ruxton & Beauchamp, 2008). The relevant data sets were examined to ensure that the correct statistical
and pairwise post hoc tests were applied (Welch, 1947; Massey, 1951; Dunn, 1964; Maxwell & Delaney,
2004; Steinskog et al., 2007; Sawyer, 2009; McKnight et al., 2010; Moder, 2010; Witte & Witte, 2017;
Delacre et al., 2019). Determining which tests were applied for a specific factor was based on the sample
coming from a population of a specific distribution, then verifying heterogeneity or homogeneity of variances.
This established the following hypothesis and subsequent post hoc tests, if applicable. Not all the tests were
used but were established in case of varying distributions and homogeneity or heterogeneity of variances.
Data was grouped by aggregating individual observations for a specific controlling factor. For example, the
deposition bar area in response to sediment concentration would have 3 groups, a mean area for each of
the 3 tested sediment concentrations; for the confluence angle, the bar area can only have 2 mean values
1 from each angle, so there are only 2 groups.

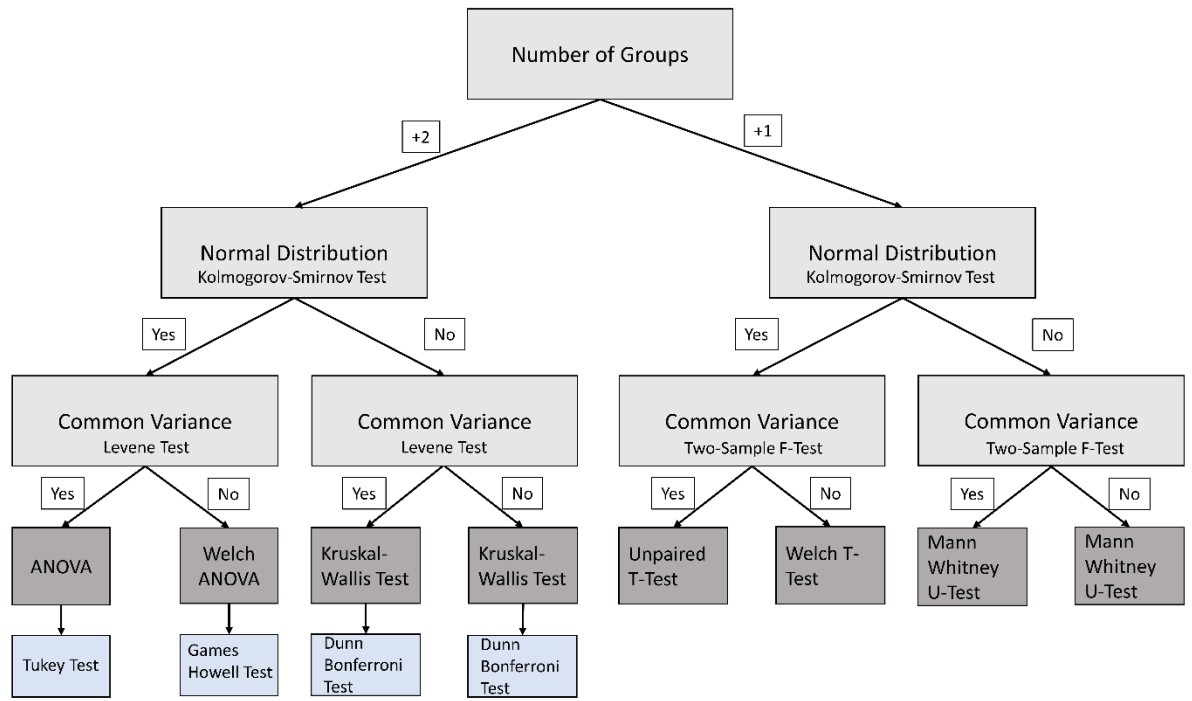


**Figure 2** Workflow for assessing the impact of controlling factors with associated tests based on the number
of groups, and the distributions and variances of the examined data sets.

**2.3    Volumetric grain sampling**

Volume samples were taken after each experiment with sample locations corresponding to confluence
morphologic (Best, 1988) and hydraulic zones (Best, 1987) in the channel. In total 8 samples were taken
for each experiment. The sampled volume was 0.002 m$^3$ with an average sample mass of 3.3 kg which was
taken by inserting a cylinder (0.16 m diameter and 0.1 m height) into the channel bed or depositional form.
The sampled mass was within the guidelines of Bunte and Abt (2001) (Eq. 2):

$$\text{Mass}_{sample}\ (kg) = 0.1 * 10^b * \rho_s * D_{max}^3 \qquad \text{(Equation 2)}$$


Where $D_{max}$ is the maximum grain size (16 mm), $\rho_s$ is grain density (2650 kg m$^{-3}$), b is the accuracy level,
high (b=5), medium (b=4), low (b=3). A larger volume would not be suitable to accurately represent small
areas of deposition or erosion as material outside of the area of interest would be additionally captured. The
samples were dried after collection and before the sieving analysis. During sieving the material was
separated into 10 fractions based on the mesh size of each sieve. The masses of each fraction were
determined and plotted as grain size distribution curves. This grain size analysis provided insights into the
hydraulic influence on the various zones.

## 271     3     Results

### 272     3.1     Development and evolution of confluence morphology


Table 4 associates the three depositional geomorphic units consistently observed for all channel
configurations and sediment concentrations with unit stream power. Unit stream power calculations are
based on initial conditions at a cross-section in the main and tributary channels. The geomorphic units were
(i) the deposition cone (Fig. 3a, Appendix 1a to 9a), (ii) transitional morphology (Fig. 3b, Appendix 1b to 9b),
and (iii) the attached-to-the-left-channel-wall separation zone bar (Fig. 3c, Appendix 1c-e to 9c-e). The scour
hole, an erosional geomorphic unit (Fig. 3), was apparent in all experiments (Appendix 1-9) on the right
bank opposite the tributary. The deposition cone was characterized by deposition upstream of the
confluence in the main channel, a compact longitudinal extent, and steep gradients in upstream and
downstream directions (Fig 3d). Cone formation resulted from insufficient transport capacity of the main
channel flow and a sustained and abundant sediment supply from the tributary channel. Deposition cones
formed for all configurations and sediment concentrations when the discharge was 15 l s$^{-1}$ and 1.5 l s$^{-1}$ in
the main and tributary channels, respectively. The transitional morphology is derived from increased
discharge and subsequent unit stream power where experimental discharges of 45 l s$^{-1}$ in the main and 4.5
l s$^{-1}$ in the tributary had nearly forced the bar over to the left bank, but morphological aspects of the
deposition cone remained. The transitional morphology partially occupies the separation zone, which is
shown in Fig. 3e where the longitudinal profile is a hybrid between the cone and bar. Discharges and related
unit stream power above 45 l s$^{-1}$ in the main and 4.5 l s$^{-1}$ in the tributary allowed for the development of an
attached-to-the-left-channel-wall separation zone bar. The bar had the greatest longitudinal extent (Fig 3f)
and the largest storage capacity for tributary-transported sediment. Once the separation zone bar was fully
developed, the hydraulic separation zone was filled with deposited sediment and flanked by the maximum
velocity zone on the right, which has been observed at lowland confluences with subcritical flows and larger
discharge ratios (Best, 1988; Biron et al., 1993; De Serres et al., 1999).

**Table 4** Geomorphic units and unit stream power ($\omega$) values. Unit stream power was calculated for the
main, tributary, and combined channel discharges. The subscripts *m* and *t* denote main and tributary
channel conditions, respectively while *tot* represents the unit stream power from the combined channel
discharge.

| EXP | $\omega_m$ | $\omega_t$ | $\omega_{tot}$ | EXP | $\omega_m$ | $\omega_t$ | $\omega_{tot}$ | EXP | $\omega_m$ | $\omega_t$ | $\omega_{tot}$ | Geomorphic Unit |
|---|---|---|---|---|---|---|---|---|---|---|---|---|
| [-] | | [W m$^{-2}$] | | [-] | | [W m$^{-2}$] | | [-] | | [W m$^{-2}$] | | [-] |
| 1 | 0.8 | 7.5 | 0.8 | 16 | 0.8 | 3.4 | 0.9 | 31 | 0.7 | 7.8 | 0.8 | Deposition cone |
| 2 | 2.2 | 21.3 | 2.5 | 17 | 2.3 | 11 | 2.5 | 32 | 2.2 | 21.2 | 2.4 | Transitional |
| 3 | 3.7 | 36.4 | 4.1 | 18 | 3.7 | 18.6 | 4.1 | 33 | 3.7 | 37.6 | 4.1 | Attached-to-channel bar |
| 4 | 5.1 | 51.9 | 5.7 | 19 | 5.1 | 25.6 | 5.6 | 34 | 5.2 | 51.3 | 5.7 | Attached-to-channel bar |
| 5 | 6.6 | 65.9 | 7.3 | 20 | 6.6 | 33.2 | 7.3 | 35 | 6.7 | 66.2 | 7.3 | Attached-to-channel bar |
| 6 | 0.7 | 7.2 | 0.8 | 21 | 0.8 | 3.5 | 0.8 | 36 | 0.7 | 7.5 | 0.8 | Deposition cone |
| 7 | 2.3 | 21.7 | 2.5 | 22 | 2.3 | 10.6 | 2.5 | 37 | 2.2 | 21.8 | 2.5 | Transitional |
| 8 | 3.7 | 36.6 | 4.1 | 23 | 3.7 | 18.3 | 4.0 | 38 | 3.7 | 36.8 | 4.1 | Attached-to-channel bar |
| 9 | 5.2 | 51.4 | 5.7 | 24 | 5.2 | 25.6 | 5.7 | 39 | 5.2 | 51.4 | 5.7 | Attached-to-channel bar |
| 10 | 6.6 | 65.8 | 7.3 | 25 | 6.6 | 32.9 | 7.3 | 40 | 6.7 | 65.7 | 7.3 | Attached-to-channel bar |
| 11 | 0.7 | 7.4 | 0.8 | 26 | 0.7 | 3.8 | 0.8 | 41 | 0.7 | 7.0 | 0.8 | Deposition cone |
| 12 | 2.2 | 22.4 | 2.4 | 27 | 2.1 | 10.9 | 2.4 | 42 | 2.2 | 21.4 | 2.4 | Transitional |
| 13 | 3.7 | 37.5 | 4.1 | 28 | 3.7 | 18.7 | 4.1 | 43 | 3.7 | 37.4 | 4.1 | Attached-to-channel bar |
| 14 | 5.2 | 51.2 | 5.7 | 29 | 5.2 | 25.7 | 5.7 | 44 | 5.2 | 51.1 | 5.7 | Attached-to-channel bar |
| 15 | 6.6 | 66.6 | 7.3 | 30 | - | - | - | 45 | 6.6 | 66.1 | 7.3 | Attached-to-channel bar |


The scour hole was created hydraulically by the extent of the separation zone forcing the confluent streams
to a smaller area, and physically by channel constriction resulting from depositional patterns reducing the

area in which the confluent flows may travel (Guillén-Ludeña et al., 2015; St. Pierre Ostrander et al., 2023), thereby increasing flow velocities (Rhoads and Kenworthy, 1995) and transport capacities. Additionally, the absence of avalanche faces inhibits the development of lee-side flow separation cells (Roy & Bergeron, 1990), which segregates sediment around the confluence instead of through it. Field observation of a gravel-bed confluence showed that tracked particles from both channels converge towards the scour hole with no noticeable segregation (Roy & Bergeron,1990). As the hydraulic separation zone filled with sediment, the spatial extent of the scour hole increased. The system tended towards an equilibrium state where sediment was transported through the scour hole, as this was the only available pathway through the confluence. The size and depth of the scour hole were greatest at lower sediment concentrations, given the same discharge. There was less sediment to be transported and potentially deposited in the scour hole, and the transport capacity of the main channel was not yet exhausted.

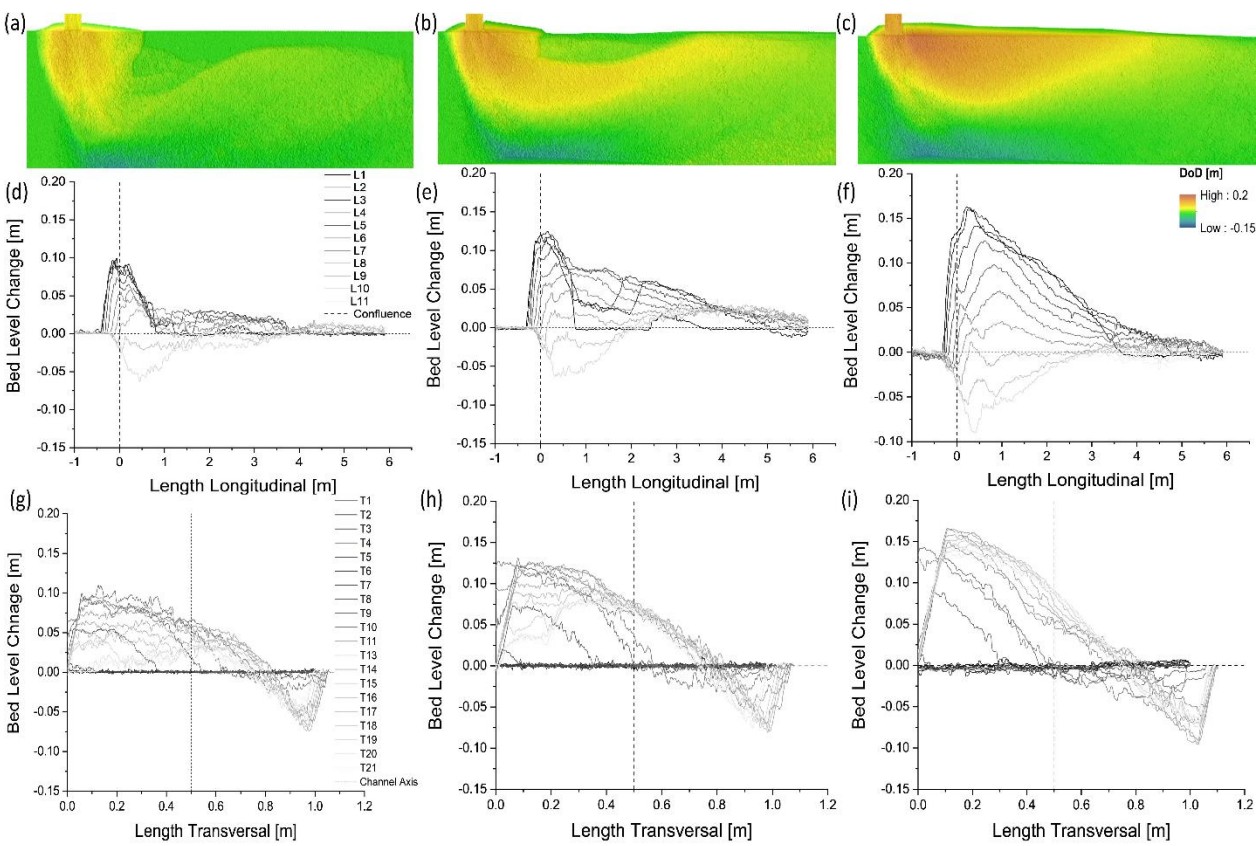

**Figure 3** Observed geomorphic units, the deposition cone (a) shown with longitudinal (d) and transversal plots (g), the transitional morphology (b) shown with longitudinal (e) and transversal plots (h), and the attached-to-channel-wall separation zone bar (c) shown with longitudinal (f) and transversal plots (i) with

the scour hole on the right, opposite the tributary. Longitudinal profiles were spaced every 0.1 m and
spanned 7 m, starting 1 m upstream of the confluence, transversal profiles were spaced every 0.1 m,
starting 1 m upstream of the confluence, and spanned 2 m, focusing on the confluence zone.

**3.2    Effects of the tributary gradient**

Figure 4 shows the DoDs from the minimum (Fig 4a, d, g), median (Fig 4b, e, h), and maximum (Fig 4c, f)
experimental discharge combinations which were produced by subtracting the DoDs from experiments 16-
30, with a 5% tributary gradient from experiments 1-15, with a 10% tributary gradient. The same general
morphological patterns consistently occurred regardless of the imposed geometric change. Intense bedload
transport in the tributary provided an abundance of sediment to the confluence. A smaller tributary gradient
of 5% (EXP 16-30) led to reduced velocity and subsequent transport capacity which did not greatly impact
the morphological development of the confluence, relative to the depositional forms observed when the
gradient was 10% (EXP 1-15). This trend could be associated with the unit stream power of the main
channel since the same patterns were observed for all sediment concentrations. As described by Guillén-
Ludeña et al. (2017), the main channel supplies the dominant flow at mountain river confluences, if the flow
is unchanged then similar development occurs. Main channel unit stream power was consistent for all
comparable experiments, the tributary unit stream power was approximately halved when the channel
gradient was reduced to 5% (EXP 16-30) (Table 4).

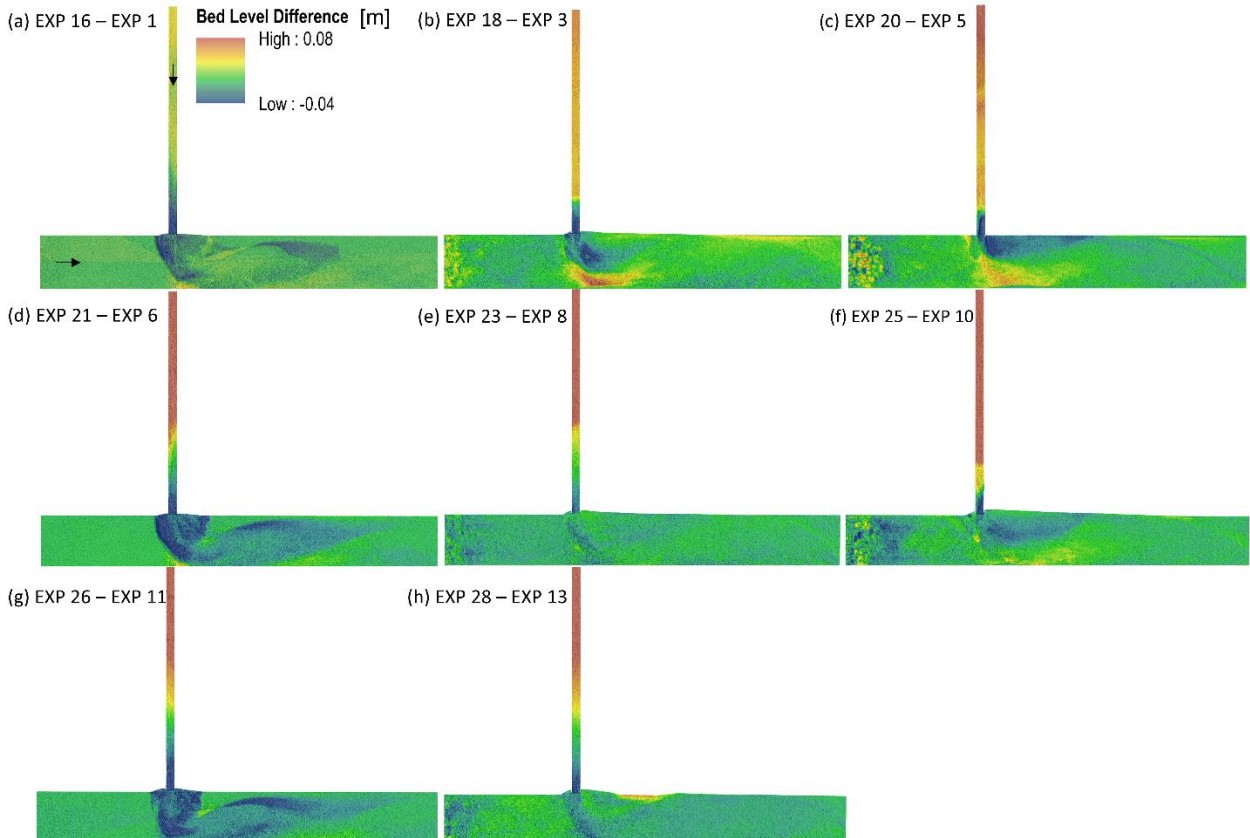

Figure 4 DoDs showing the morphological differences between the minimum (a ,d, g), median (b, e, h), and maximum (c, f) experimental discharges which were created by subtracting the DoDs from experiments with a 5% tributary gradient (EXP 16-30) from the DoDs with a 10% tributary gradient (EXP 1-15).

Figure 5 shows the depositional and erosional characteristics of experiments 1-15 (10% tributary gradient, 90° confluence angle) and 16-30 (5% tributary, 90° confluence angle) excluding the tributary channel. A visual inspection of Fig. 5 does not show a clear trend in differences in depositional or erosional characteristics between gradients. What trend could be inferred is most apparent when comparing the first 5 experiments for each geometry group (EXP 1-5 and EXP 16-20). Depositional patterns (Fig. 5a, 5c, and 5e) were greater for experiments 16-20 than for experiments 1-5, while erosional patterns were greater for experiments 1-5 than for 16-20 (Fig. 5b, 5d, and 5f). Reducing the tributary channel gradient reduced the velocity of the tributary flow (Table 1), limiting its contribution to main channel erosion. When the tributary gradient was 10% (EXP 1-15), there was greater penetration of the tributary flow into the main channel and

a local increase in transport capacity, creating a larger and deeper scour hole and enhanced conveyance
of sediment through the confluence.

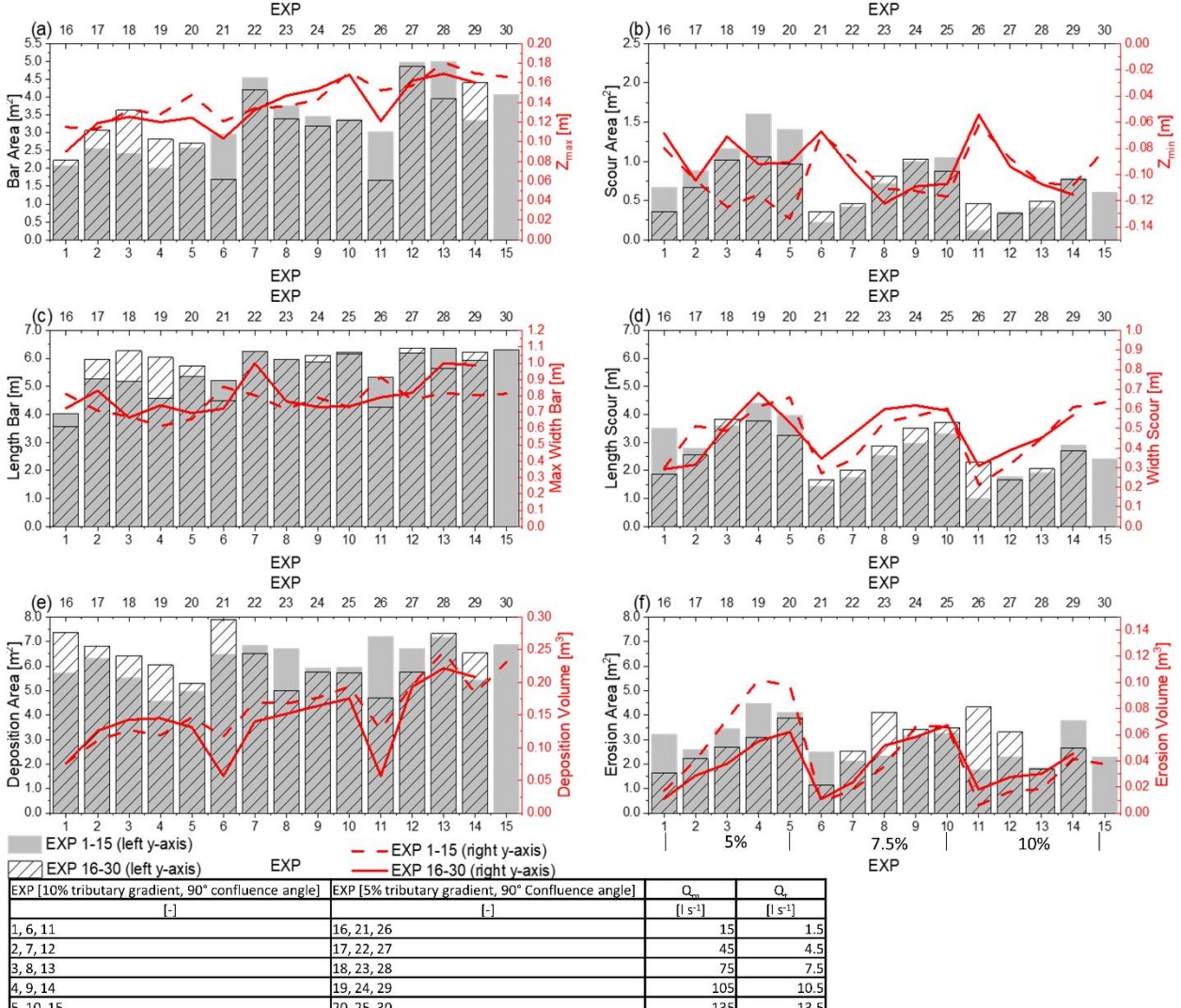


**Figure 5** A comparison of morphological attributes across experiments with a 5% (EXP 16-30) and 10%
tributary gradient (EXP 1-15), sediment concentration groups are shown in panel f. Deposition bar and scour
areas (a, b) are delineated by deposition or erosion above or below 0.01 m, respectively. The width and
length values represent the maximum measured width or length (c, d), while the main channel deposition
and erosion areas (e, f) represent all deposition and erosion in the main channel.

Figure 6 shows the gradients and volumes of the deposited sediment in the tributary channel at the end of
experiments 1-30. The depositional gradient was determined through a linear regression of the DoD surface
profile of the tributary channel. Adjustments to the tributary gradient changed the depositional mechanisms
in the tributary channel, characterized by either an increase or decrease in the gradient of the deposited
material in the tributary channel, relative to the initial gradient. When the initial gradient was 10% (EXP 1-
15), the transport capacity of the main channel was the limiting factor for sediment moving through the
confluence. This led to a regressive aggradation of sediment, starting at the junction, which decreased the
gradient of the tributary channel. Conversely, when the initial tributary channel gradient was 5% (EXP 16-
30), the resulting decrease in velocity saturated the transport capacity of the tributary channel.
Consequently, the depositional patterns switched, and intense progressive deposition occurred starting at
the upstream boundary of the tributary channel which increased the gradient of the channel.

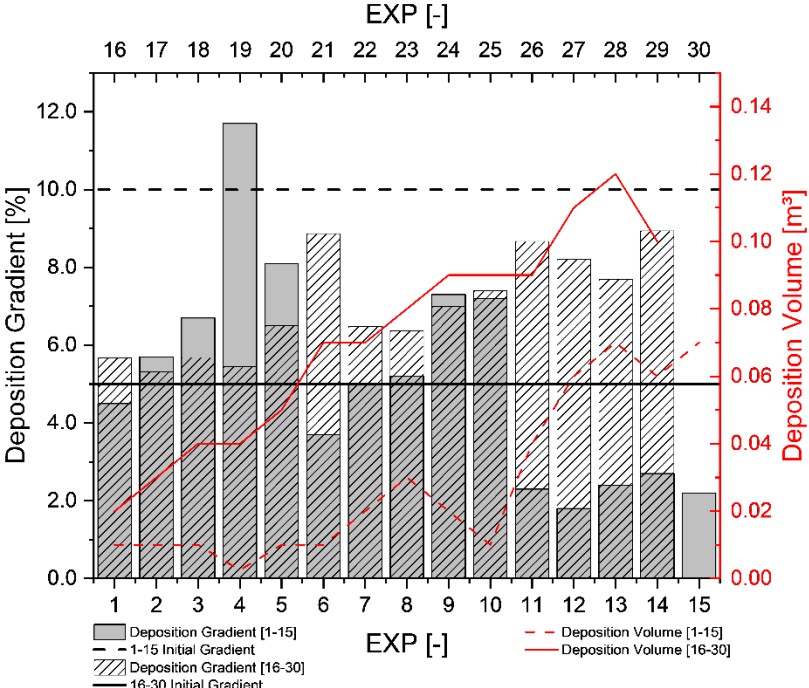


**Figure 6** Gradients and volumes of deposited sediment in the tributary channel for experiments 1-15 with
an initial 10% tributary gradient and experiments 16-30 with an initial 5% tributary gradient.

**3.3    Effects of the confluence angle**

Figure 7 shows the DoDs from the minimum (Fig 7a, d, g), median (Fig 7b, e, h), and maximum (Fig 7c, f, i) experimental discharge combinations which were created by subtracting the DoDs produced from experiments with a 45° confluence angle (EXP 31-45) from the DoDs with a 90° confluence angle (EXP 1-15). The tributary channels with a 45° confluence angle were extracted and referenced to the 90° tributary channels allowing for DoD comparisons. A visual inspection of confluence zone morphology does not reveal drastic changes between confluence angle experiments. Small regions of morphological change are apparent, mainly increased deposition downstream of the junction corner and a generally shallower scour hole when the confluence angle was 45°.

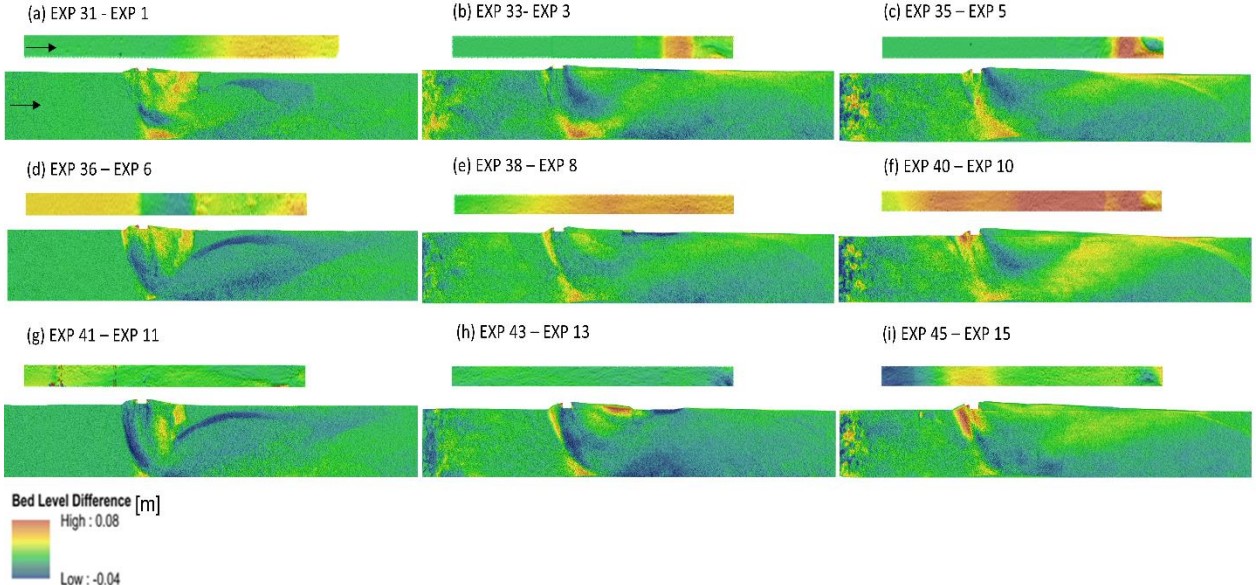

**Figure 7** DoDs showing the morphological differences between the minimum (a, d, g), median (b, e, h), and maximum (c, f, i) experimental discharges which were created by subtracting the DoDs from experiments with a 45° confluence angle (EXP 31-45) from the DoDs with a 90° confluence angle (EXP 1-15).

Figure 8 shows subtle morphological differences with noticeable trends of scour characteristics, while depositional characteristics do not exhibit standout trends upon visual assessment. Both the area and length of the scour hole tended to be greater for experiments 31-45, with a 45° confluence angle (Fig. 8b and 8d).

However, the depth of scour and width of the scour was generally greater for experiments 1-15, with a 90°
confluence angle. For both confluence angle experiment groups, a clear trend of increasing scour area,
length of scour, and erosion area occurred within each sediment concentration group, increasing in
response to discharge. Assessing the impact of confluence angle adjustments on depositional attributes
required a statistical approach to reveal any nuanced relationships occurring within the channel.

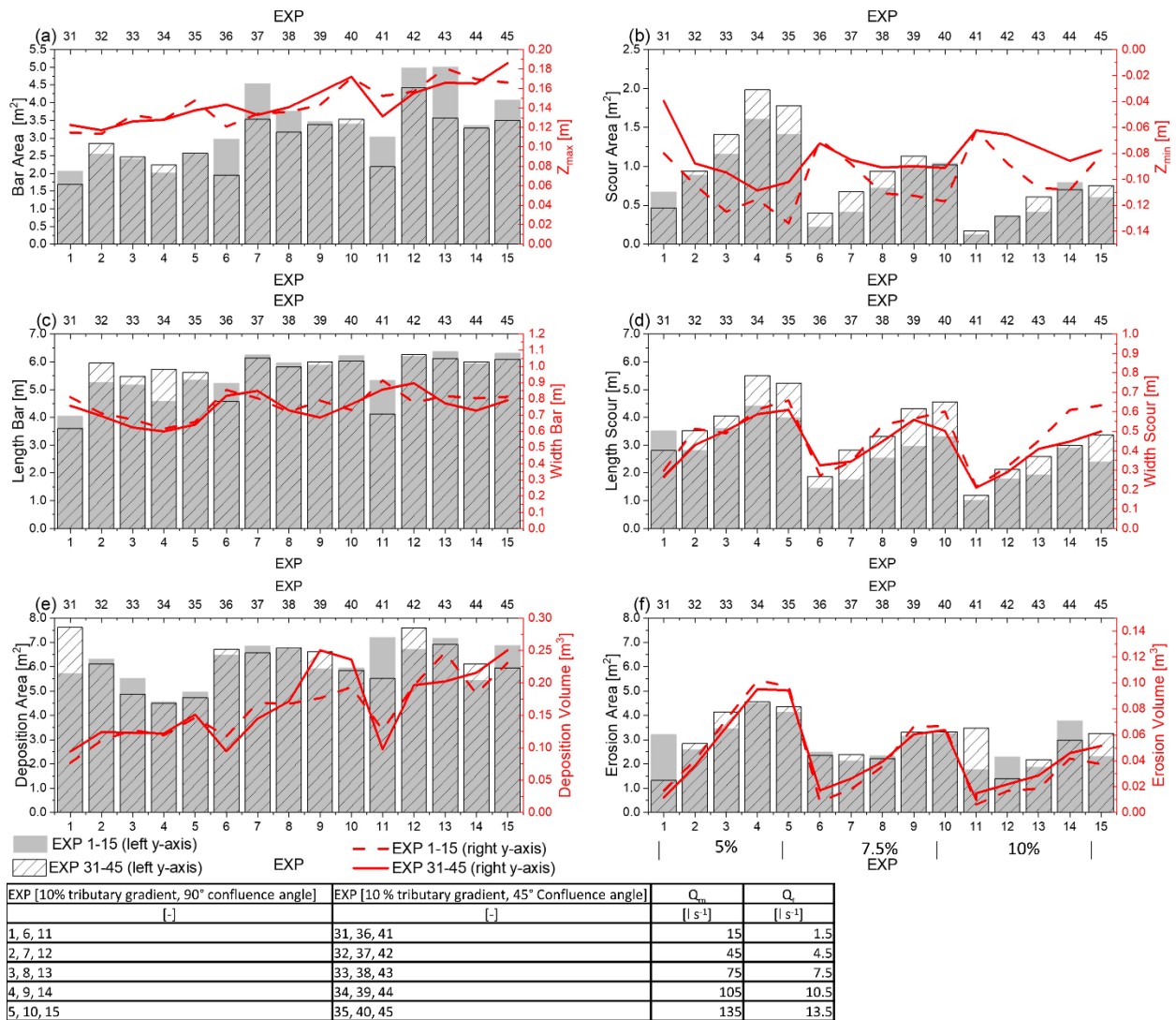

| EXP [10% tributary gradient, 90° confluence angle] | EXP [10 % tributary gradient, 45° Confluence angle] | $Q_m$ | $Q_t$ |
|---|---|---|---|
| [-] | [-] | [l s⁻¹] | [l s⁻¹] |
| 1, 6, 11 | 31, 36, 41 | 15 | 1.5 |
| 2, 7, 12 | 32, 37, 42 | 45 | 4.5 |
| 3, 8, 13 | 33, 38, 43 | 75 | 7.5 |
| 4, 9, 14 | 34, 39, 44 | 105 | 10.5 |
| 5, 10, 15 | 35, 40, 45 | 135 | 13.5 |


**Figure 8** Comparison of morphological attributes across experiments with a 45° confluence angle (EXP 31-
45) and experiments with a 90° confluence angle (EXP 1-15). Deposition bar and scour areas (a, b) are
delineated by deposition or erosion above or below 0.01 m, respectively. The width and length values
represent the maximum measured width or length (c, d), while the main channel deposition and erosion
areas (e, f) represent all deposition and erosion in the main channel.
Figure 9 illustrates that variations in tributary depositional properties occurred despite maintaining a
consistent tributary gradient across the experimental groups. When the confluence angle was 45° (EXP 31-
45), a near overall increase in the depositional volume and a decrease in the depositional gradient was
observed (Fig. 9) relative to experiments with a 90° confluence angle (EXP 1-15). A reduction in the
confluence angle limits tributary channel flow penetration into the main channel (Best, 1988), reducing the
exposure of the tributary sediment to main channel entraining forces. In the context of experiments 1-15,
with a greater confluence angle (90°), the penetration of the tributary channel exhibited a greater extent.
Increasing the confluence angle caused a greater mutual deflection of flows, further segregating the
tributary and main channel flows (Best, 1987). This factor, coupled with the increased velocity, allowed the
tributary sediment load to rapidly pass through the confluence zone when the confluence angle was 90°
rather than be deposited in the tributary channel.

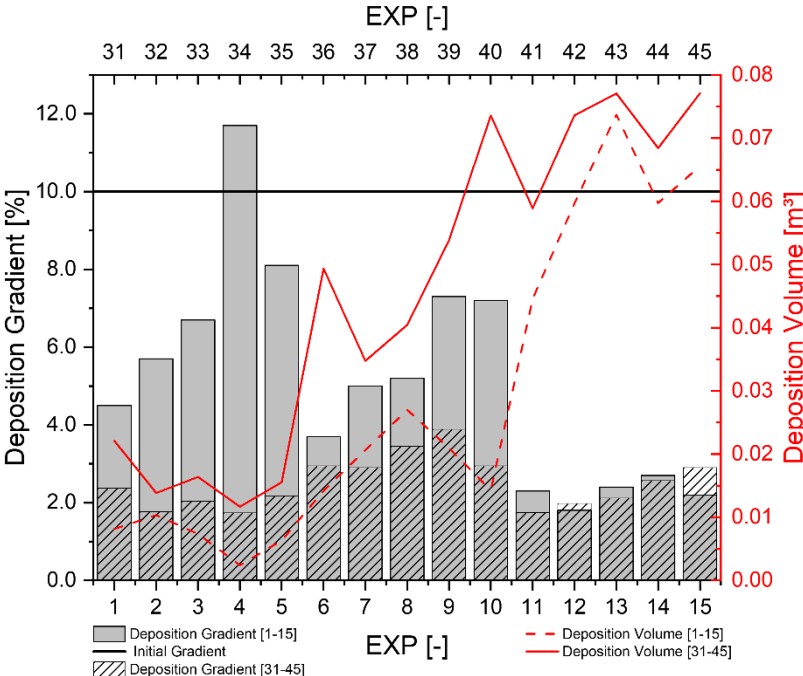


**Figure 9** Gradients and volumes of deposited sediment in the tributary channel for experiments 1-15 (10%
tributary gradient, 90° confluence angle) and 31-45 (10% tributary gradient, 45° confluence angle).
**3.4    Statistical analysis of controlling factors impacting confluence morphology**
**3.4.1    Overview**

Only controlling factors that had a significant effect (Table 5) on the response variables of the main channel
are discussed. The focus of the statistical analysis was to determine the dominant controls over confluence
morphology. For this reason, tributary channel depositional behavior was not included as a response
variable.

**Table 5** Introduced controlling factors and their impact on confluence morphology, bold text indicates the
factor had a significant impact on one or more groups of the response variable. P-values from overall mean
comparison tests are included.

| Factor | $z_{max}$ | $z_{min}$ | Deposition area | Deposition volume | Erosion area | Erosion volume | Bar area | Bar length | Bar width | Scour area | Scour length | Scour width |
|---|---|---|---|---|---|---|---|---|---|---|---|---|
| Sediment concentration | **<.0001** | .30 | .09 | **.001** | .19 | **.015** | **2.85E-4** | .059 | **<.0001** | **4.38E-4** | **3.63E-4** | .30 |
| Discharge | **.004** | **<.0001** | **.047** | **<.0001** | **.007** | **<.0001** | **1.89E-4** | **<.0001** | .14 | **<.0001** | **<.0001** | **<.0001** |
| Tributary gradient | .20 | .78 | .82 | .24 | .96 | .50 | .27 | .79 | .21 | .33 | .35 | .55 |
| Confluence angle | .46 | **0.022** | .91 | .40 | 0.84 | .67 | .25 | .81 | .37 | .23 | **.047** | .267 |


**3.4.2    Sediment concentration**

Table 6 and Fig. 10 show that sediment concentration had a significant impact on 7 out of 12 response
variables. Increased or decreased sediment concentration enhanced depositional or erosional patterns,
respectively. Post hoc testing further revealed patterns caused by the sediment concentration (Table 6).
Unsurprisingly, the majority of the significant differences in mean response values occurred between 5%
and 10% sediment concentration groups. The maximum deposition depth was significantly reactive to all
sediment concentrations. With increasing sediment concentration the deposition depth increased but
reached a maximum as aggradation cannot exceed the local flow depth. When the sediment concentration
was 7.5%, the response variables did not significantly differ from those of the 5% and 10% groups.
**Table 6** Sediment concentration and its impact on the response variables; (σ) is the standard deviation.
Pairwise post hoc mean comparison testing is summarized with letters A, B, and C. Means that do not share
a letter are significantly different. For example, the mean $Z_{max}$ for each sediment concentration group was
significantly different (A, B, C), but the mean deposition volume for 7.5% and 10% sediment concentration
groups did not significantly differ from each other (B, B) but were significantly different from the mean
deposition volume when the sediment concentration was 5% (A).

| Response Variable | σ | | | Test | Difference in Means | Post hoc Test | 5 | 7.5 | 10 |
|---|---|---|---|---|---|---|---|---|---|
| | 5% | 7.5% | 10% | | | | | | |
| [-] | [-] | [-] | [-] | [-] | [-] | [-] | [%] | [%] | [%] |
| $Z_{max}$ [m] | 0.01 | 0.02 | 0.02 | ANOVA (F = 18.5) | Yes | Tukey-Test | A | B | C |
| $Z_{min}$ [m] | 0.02 | 0.02 | 0.02 | ANOVA (F = 1.2) | No | | | | |
| Deposition area [m²] | 1.00 | 0.68 | 0.85 | ANOVA (F = 2.4) | No | | | | |
| Deposition volume [m³] | 0.02 | 0.05 | 0.06 | ANOVA (F = 8.2) | Yes | Tukey-Test | A | B | B |
| Erosion area [m²] | 1.02 | 0.74 | 0.87 | ANOVA (F = 1.7) | No | | | | |
| Erosion volume [m³] | 0.03 | 0.02 | 0.01 | Welch ANOVA (F = 4.9) | Yes | Games-Howell | A | A/B | B |
| Deposition bar area [m²] | 0.47 | 0.72 | 1.01 | Welch ANOVA (F = 11.5) | Yes | Games-Howell | A | B | B |
| Length bar [m] | 0.88 | 0.57 | 0.74 | ANOVA (F = 3.0) | No | | | | |
| Width bar [m] | 0.07 | 0.08 | 0.09 | ANOVA (F = 13.3) | Yes | Tukey-Test | A | B | B |
| Scour area [m²] | 0.47 | 0.30 | 0.22 | Welch ANOVA (F= 10.6) | Yes | Games-Howell | A | A | B |
| Length scour [m] | 0.96 | 0.96 | 0.67 | ANOVA (F = 9.7) | Yes | Tukey-Test | A | B | B |
| Width scour [m] | 0.14 | 0.12 | 0.14 | ANOVA (F = 1.3) | No | | | | |


Adjustments in deposition and erosion areas allowed for the majority of the incoming sediment load to pass
through the confluence. However, given the differences in sediment loads, rapid mutual adjustments were
morphologically represented by the same general patterns but with less erosion and more aggradation as
sediment concentration increased. The differences in mean response values between the experiments with
5% and 10% tributary sediment concentrations and the similarities to the mean response values, when the
sediment concentration was 7.5%, can be attributed to this process.

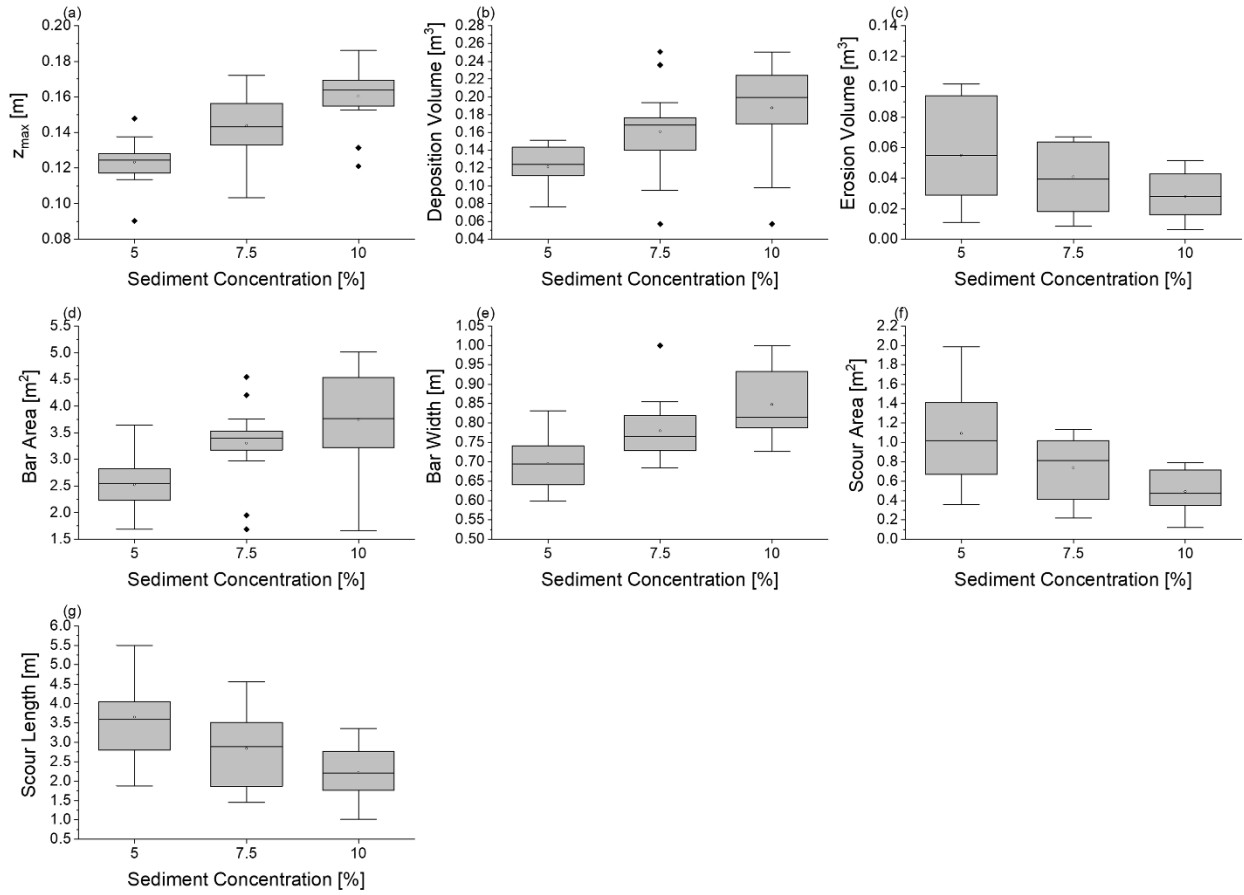

**Figure 10** Boxplots from ANOVA and Welch ANOVA results for all response variables that showed a significant difference in mean values (Table 5) with sediment concentration as the controlling factor.

### 3.4.3 Combined discharge

Table 7 and Fig. 11 show that the discharge significantly affected 11 out of 12 response variables. Generally, erosional processes increased with increasing discharge as the transport capacity of the main channel flow increased. At lower discharges with limited transport capacity, erosional processes were comparatively reduced. However, certain instances revealed increased depositional properties with increasing discharge (Fig. 11a and 11d). This most apparently occurred between the 16.5 l s$^{-1}$ and 49.5 l s$^{-1}$ combined discharge experiments. A deposition cone formed across all sediment concentrations when the combined discharge was 16.5 l s$^{-1}$. Unlike the bar or transitional morphology, the deposition cone does not occupy the separation

zone and is characterized by a short longitudinal extent while protruding furthest into the main channel from
the tributary channel. At discharges at and above 49.5 l s$^{-1}$, the depositional patterns shifted, and sediment
was entrained and deposited in the separation zone. The separation zone is the largest sink for tributary-
transported sediment; the occupying bar can only be as big as the hydraulic zone, which is the same size
for a given discharge ratio (Best, 1987; 1988). This explains the subtle differences in depositional properties
once the combined discharge exceeded 49.5 l s$^{-1}$.

**Table 7** Discharge and its impact on the response variables; ($\sigma$) is the standard deviation. Pairwise post
hoc mean comparison testing is summarized with letters A, B, C, and D; means that do not share a letter
are significantly different.

| Response Variable | σ | | | | | Test | Diff. in means | Post Hoc Test | 16.5 | 49.5 | 82.5 | 116 | 149 |
|---|---|---|---|---|---|---|---|---|---|---|---|---|---|
| | 16.5 | 49.5 | 82.5 | 115.5 | 148.5 | | | | | | | | |
| [-] | [l s$^{-1}$] | [l s$^{-1}$] | [l s$^{-1}$] | [l s$^{-1}$] | [l s$^{-1}$] | [-] | [-] | [-] | [l s$^{-1}$] | [l s$^{-1}$] | [l s$^{-1}$] | [l s$^{-1}$] | [l s$^{-1}$] |
| $Z_{max}$ [m] | 0.02 | 0.02 | 0.02 | 0.02 | 0.02 | ANOVA (F = 4.5) | YES | Tukey-Test | A | A/B | A/B | A/B | B |
| $Z_{min}$ [m] | 0.01 | 0.01 | 0.02 | 0.01 | 0.02 | ANOVA (F = 10.7) | YES | Tukey-Test | A | B | B | B | B |
| Deposition [m$^2$] | 1.07 | 0.52 | 0.93 | 0.77 | 0.68 | ANOVA (F = 2.7) | YES | Tukey-Test | A | A | A | A | A |
| Deposition [m$^3$] | 0.02 | 0.03 | 0.04 | 0.04 | 0.05 | ANOVA ( F = 9.3) | YES | Tukey Test | A | B | B | B | B |
| Erosion area [m$^2$] | 1.08 | 0.52 | 0.92 | 0.66 | 0.63 | ANOVA (F = 4.1) | YES | Tukey Test | A | A | A/B | B | A/B |
| Erosion volume [m$^3$] | 0.004 | 0.01 | 0.02 | 0.02 | 0.02 | Welch ANOVA (F = 28.9) | YES | Games-Howell | A | B | B/C | C | C |
| Bar area [m$^2$] | 0.52 | 0.91 | 0.79 | 0.71 | 0.54 | ANOVA (F= 7.2) | YES | Tukey Test | A | B | B | B | B |
| Length bar [m] | 0.62 | 0.33 | 0.38 | 0.5 | 0.34 | ANOVA (F = 22.0) | YES | Tukey Test | A | B | B | B | B |
| Width bar [m] | 0.06 | 0.11 | 0.11 | 0.12 | 0.06 | ANOVA (F = 1.9) | NO | | | | | | |
| Scour area [m$^2$] | 0.17 | 0.24 | 0.33 | 0.42 | 0.38 | ANOVA (F = 9.1) | YES | Tukey Test | A | A/B | B/C | C | C |
| Length scour [m] | 0.8 | 0.63 | 0.76 | 0.92 | 0.87 | ANOVA F = 8.4) | YES | Tukey Test | A | A | A/B | B | B |
| Width scour [m] | 0.05 | 0.08 | 0.06 | 0.06 | 0.06 | ANOVA (F = 36.9) | YES | Tukey Test | A | B | C | D | D |


Pair-wise post hoc comparisons of maximum deposition depth indicated a significant difference in mean
values between the lowest and highest combined discharge experiments while revealing similarities among
intermediate discharge scenarios. These similarities could be attributed to the combined flows regulating
the depositional depth, which does not exceed the flow depth. The observed differences can be attributed
to the increased sediment load and associated morphological changes with increasing discharge.

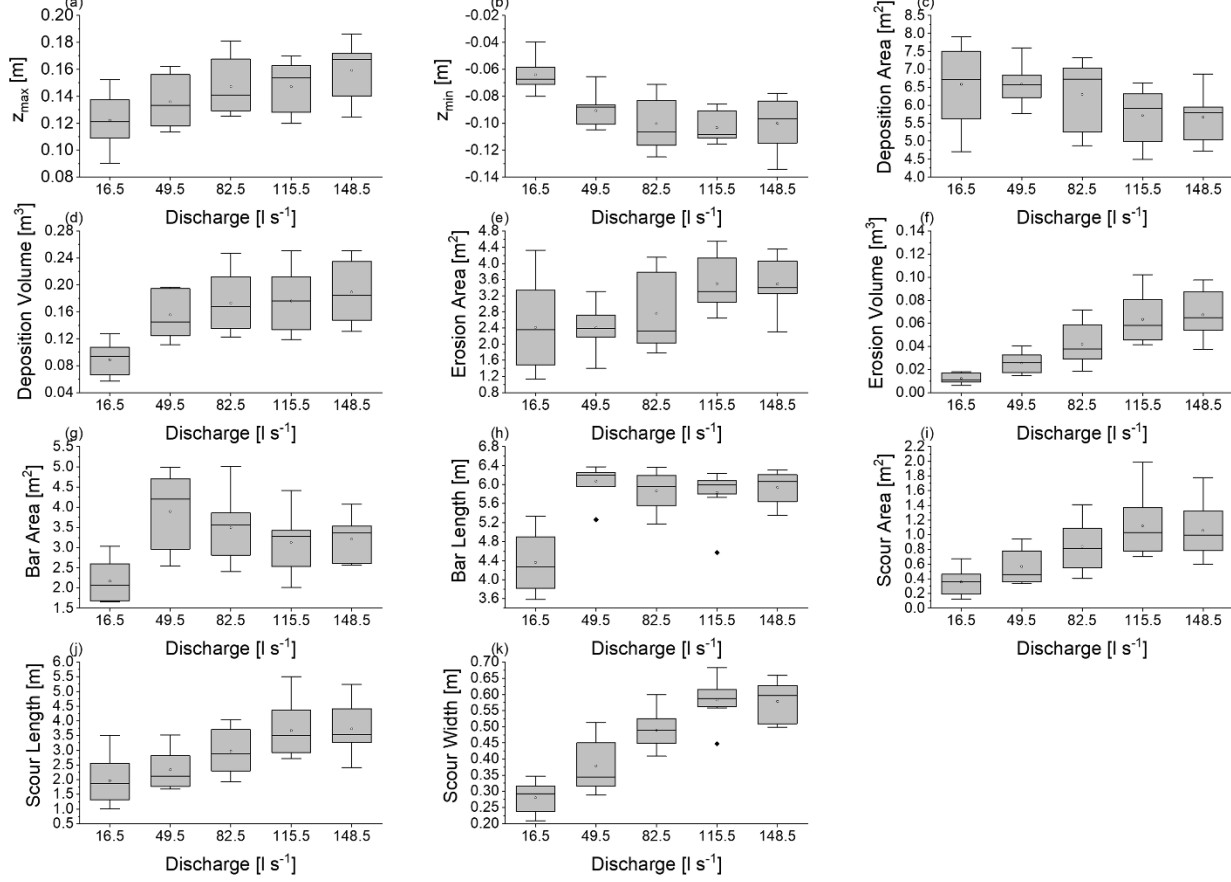

**Figure 11** Boxplots from ANOVA and Welch ANOVA results for all response variables that showed a significant difference in mean values (Table 5) with combined discharge as the controlling factor.

### 3.4.4 Confluence angle

Surprisingly, the confluence angle only had a significant influence on 2 out of the 12 response variables (Table 8). The confluence angle did have a decisive impact on scour depth (Fig. 12a). This could be attributed to the degree of turbulence increasing with increasing confluence angle (Mosley, 1976). The elevated turbulence arises from the increased mutual flow deflection, which influences the shear layers generated between the two converging flows. Along these shear layers, powerful vortices are created which enhance the bed shear stress within the junction, resulting in significant bed scour (Best, 1987). Reducing the confluence angle allowed for improved mixing of tributary and main channel flows, which in turn decreased the turbulence in the confluence producing shallower scour.

**Table 8** Confluence angle and its impact on the response variables. Post hoc testing was not required since there are only two groups to compare; σ is the standard deviation.

| | σ | | | |
| Response Variable | 45° | 90° | Test | Difference in means |
| [-] | [-] | [-] | [-] | [-] |
| $Z_{max}$ [m] | 0.02 | 0.02 | T-Test (t statistic = - 0.742) | NO |
| $Z_{min}$ [m] | 0.02 | 0.02 | T Test ( t statistic = -2.37) | YES |
| Deposition Area [m²] | 0.96 | 0.85 | T Test (t statistic = 0.109) | NO |
| Deposition Volume [m³] | 0.06 | 0.05 | T Test (t statistic = -0.843) | NO |
| Erosion Area [m²] | 0.98 | 0.87 | T Test (t statistic = -0.199) | NO |
| Erosion Volume [m³] | 0.03 | 0.03 | T Test (t statistic = -0.425) | NO |
| Deposition Bar Area [m²] | 0.75 | 0.95 | T Test (t statistic = 1.169) | NO |
| Length Bar [m] | 0.81 | 0.77 | T Test (t statistic = 0.238) | NO |
| Width Bar [m] | 0.10 | 0.10 | T Test (t statistic = 0.916) | NO |
| Scour Area [m²] | 0.52 | 0.36 | T Test (t statistic = -1.212) | NO |
| Length Scour [m] | 1.22 | 0.88 | T Test (t statistic = -2.04) | YES |
| Width Scour [m] | 0.12 | 0.14 | T Test (t statistic = 1.125) | NO |

Additionally, the confluence angle had an impact on the length of the scour (Fig. 12b). Enhanced mixing of confluent flows and a reduced hydraulic separation zone created conditions where the scour generally occupied a greater area but produced a shallower scour depth. However, the width of the bar was relatively unchanged (Fig. 8c) in response to the confluence angle; the increased scour area was represented by an increase in scour length. While the penetration of the tributary channel was reduced, the transport capacity of the main channel was still sufficient to mobilize a similar volume of sediment (Fig. 8f).

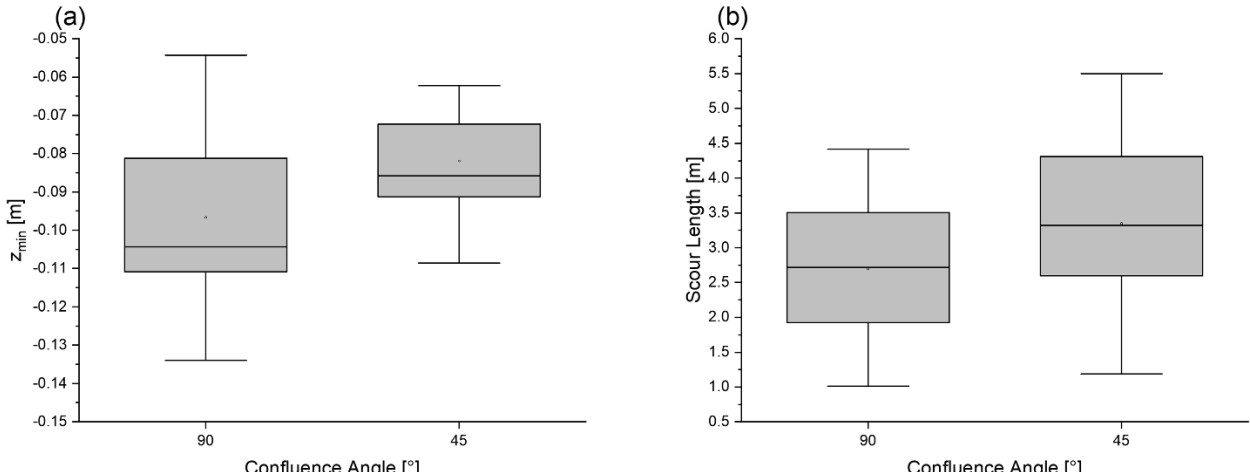

**Figure 12** Boxplots from T-Test results for all response variables that showed a significant difference in mean values (Table 5) with the confluence angle as the controlling factor.

**4    Discussion**
**4.1    Special dynamics of mountain river confluences**

The confluence angle has been established as one of the main drivers of confluence morphology, thus
affecting the spatial distribution of the hydraulic zones for lowland confluences. However, for mountain river
confluences during events with intense bedload transport it had a minimal effect, corroborating hypothesis
1, that adjustments to the confluence angle (Fig. 8, Table 8) and the tributary gradient (Fig. 5, Table 5) do
not significantly impact confluence morphology and the development of specific geomorphic units. Wohl
(2010) discusses the extremal hypotheses (Davies & Sutherland, 1983) which are based on the underlying
assumption that the equilibrium channel morphology corresponds to the morphology that maximizes or
minimizes the value of a specific parameter (Darby and Van De Wiel, 2003). Examples of this are reductions
of unit stream power (Yang & Song, 1979) and energy dissipation rate (Yang, 1976) and maximizations of
friction factor (Davies and Sutherland, 1983), and sediment transport rate (White et al., 1982). The
confluence morphologically reacted to the steep channel flooding and bedload conditions, characterized by
higher velocities, sediment concentrations, and Froude numbers than what would be expected at a lowland
confluence, and adjusted to maximize sediment transport through the confluence. Since all channel
geometry experiments were exposed to the same discharges and sediment supply rates, a similar
development occurred. Lowland regions are typically less intense and morphologically more responsive,
relative to mountain river confluences during flooding events, to variations in the size and orientation of the
hydraulic zones as they respond to channel adjustments (Mosley, 1976; Best 1987, 1988; Liu et al., 2015).
Scour area and depth were the only response variables sensitive to the confluence angle. Decreasing the
confluence angle limited the extent of the flow separation zone (Mosley, 1976; Best, 1987). The zone of
maximum velocity responded to the size of the flow separation zone (Best, 1987). When more channel was
available for the zone of maximum velocity from the decreased size of the separation zone, the velocity
decreased, causing shallower scour, which is consistent with the findings of Mosley (1976) and Best (1988).
In contrast, increasing the confluence angle increased the local velocity and transport capacity and caused
greater penetration of the tributary flow. These combined aspects provide evidence that the transport
capacity of the main channel is enhanced at higher confluence angles, which was reflected in the tributary
depositional volumes and gradients. It has been previously observed in mountain rivers (Mueller & Pitlick,
2005; Trevisani et al., 2010) that the tributary channel gradient responds to the transport capacity of the
flow. Mueller and Pitlick (2005) suggest that forced changes in gradient are offset by adjustments to width,
depth, and bed surface texture to maintain a balance between the intensity and frequency of bed load
transport. In confined channels, width adjustments are not possible, resulting in extensive deposition in the
channel. The main differences in sediment depositional patterns and mechanisms from adjusting the
tributary channel gradient were observed in the tributary channel, while the main channel was largely
unchanged. This indicates that with a sustained and abundant sediment supply and relatively uniform main
channel hydraulic conditions, the morphologic development of the confluence is not significantly impacted
by changes in the tributary channel gradient.
Referring to hypothesis 2 (sediment concentration and channel discharge exert the most control over
depositional and erosional patterns), the same geomorphic units and morphological patterns occurred for
all experimental groups and channel configurations, which establishes the dominance of the combined
channel discharge over the confluence. This can be explained according to Guillén-Ludeña et al., (2017)
where the main channel supplies the dominant flow discharge. The unit stream power in the main channel
(Table 4) was sufficient to force the development of the same geomorphic units, for a specific discharge,
regardless of changes to sediment concentration and channel geometry. Adjustments to sediment
concentration were reflected in varying ranges of deposition and erosion depths and volumes, as well as
varying extents of these geomorphic units. Interaction between discharge and sediment shows clear trends
of coarsening or fining at specific sites (Fig. 13, Appendix 10) for all the introduced controlling factors.
However, trends relating sediment concentration or channel geometry to coarsening or fining are not
apparent since the same general morphological patterns consistently occurred, which in turn caused similar
hydraulic conditions to develop. Grain size distribution curves from the tributary channel near the
confluence, the deposition cone or bar, and the recovery zone further illustrate the selective bedload
transport occurring in the confluence zone. Consistent across all experiments, the deposited material in the
tributary was finer than the input mix (Fig. 13a to 13c, Appendix 10). For experiments with the 10% tributary
gradient, this can be explained by the regressive aggradation occurring in the tributary channel, which
reduced the gradient of the tributary and, thus, its transport capacity. For experiments with a 5% tributary
gradient, the transport capacity of the tributary was saturated, which caused intense progressive deposition
of all grain sizes in the channel despite the increased depositional gradient. Samples taken from the scour
hole (Fig. 13d to 13f, Appendix 10) showed an overall coarsening, illustrating the enhanced transport
capacity through this zone. The separation zone bar was formed in a region of low flow velocity relative to
the main channel, which is reflected in the associated grain size distributions (Fig. 13h and 13i, Appendix
10). The samples taken from the lowest discharge experiments were from the deposition cone; the cone
did not occupy the hydraulic separation zone and was exposed to the main channel flow. Accordingly, the
samples showed a general coarsening pattern of the finer grain fractions and a fining of the larger grain size
fractions (Fig. 13g, Appendix 10). The zone of flow recovery is characterized by decreased turbulence and
more uniform flow patterns and bed morphology (Best, 1987; 1988). As a result, no hydraulic or morphologic
structures existed that influenced the velocity distribution throughout this portion of the channel. This is
apparent in Fig. 13j to 13l where the samples taken across all experiments showed the least deviation from
the plotted line of the input material. A slight but overall coarsening is apparent, caused by the increased
velocity from the combined channel flow and the resulting selective bedload transport.

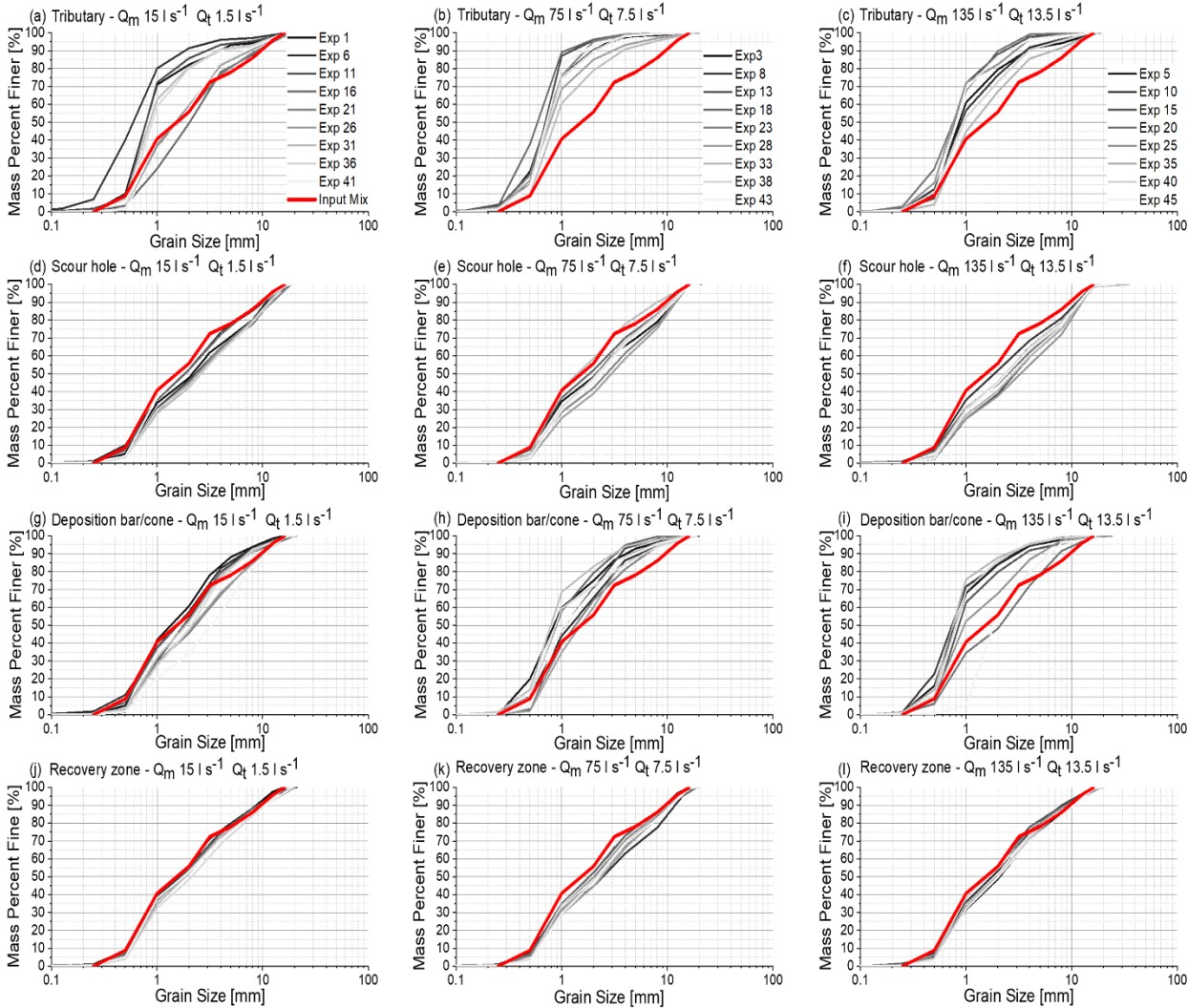

**Figure 13** Grain size distribution curves from samples taken from the tributary channel (a-c), the scour hole (d-f), the deposition cone or bar (g-i), and the recovery zone (j-l) for the lowest, middle, and highest experimental discharges, $Q_m$ and $Q_t$ denote main and tributary channel discharges, respectively.

## 4.2 Modelling limitations

Modelling limitations deal mainly with scale effects and the duration required to set up and run an experiment, limiting the scope of the study, but creating a well-founded base to build from. Preparing and running an experiment took multiple days; the project duration did not allow investigations into the effects of the discharge ratio. An ideal experimental program would have included the same 45 experiments but with a different discharge ratio. Accordingly, we strongly encourage additional investigations into this

component as it influences mountain river confluences. All physical models are subject to some degree of
scale effects as it is impossible to correctly model all force ratios (Chanson, 2004; Heller, 2011). This arises
from having to choose the most relevant force ratio, which for open channel hydraulics is Froude similarity
(Heller, 2011). Under Froude similarity, the remaining force ratios cannot be identical between model and
prototype and can result in non-negligible scale effects (Heller, 2011). Scale effects generally increase with
increasing prototype to model scale factor (Heller, 2011). Scale limitations of grain size diameters are
discussed in Zarn (1992), where grain sizes smaller than 0.22 mm can change the flow-grain interaction
due to cohesion effects. In this regard, Oliveto and Hager (2005) discuss limiting the $D_{50}$ to 0.80 mm. The
model grain size distribution has a minimum grain size of 0.5 mm and a $D_{50}$ of 1.4 mm. The Shields ($\theta$)
number and the grain Reynolds (Re*) number were calculated in the main channel for all discharges and
geometric configurations. At the lowest discharge experiments, $\theta$ and Re* at the model scale range from
0.08-0.10 and 60-67, respectively. At prototype scale Re* ranges from 9849-10927. At the next discharge
combination, $\theta$ and Re* at the model scale range 0.15-0.17 and 82-87, respectively. At prototype scale Re*
ranges from 13523-14247. While there is certainly a significant shift in Re* between lab and prototype
scales, Aufleger (2006) states that assuming Froude similarity and minimizing scale effects for pre-alpine
gravel bed rivers Re* numbers at the model scale above 80 are recommended. In this regard, for the lowest
discharge experiments, the smaller grain fractions were subject to some degree of scale effects.

**5    Conclusion**

The channel discharges and then the tributary sediment concentration are the most impactful factors
influencing mountain river confluence morphology during events with intense bedload transport. This
conclusion contrasts with the findings of the literature dealing with the controls of river confluences.
Mountain river confluences are influenced by characteristics unique to mountain regions, including the
availability of massive amounts of sediment and frequent and intense localized flooding. The rate of
sediment entering the confluence saturated the transport capacity of the main channel. The resulting
morphologies represented a system tending towards an equilibrium state, optimized to maximize sediment
transport through the confluence through local increases in sediment transport rate. Every geometric group
of experiments had the same discharges and sediment supply rates; the resulting morphologies were similar
because the channel was responding to similar intense hydraulic and sediment supply conditions. This
limited the effect the channel adjustments had on the hydraulic zones influencing confluence morphology.
However, adjustments did cause an apparent response to the depositional mechanisms in the tributary
channel. A progressive or regressive aggradation of tributary sediment occurred, which enhanced or
reduced the tributary channel transport capacity. Rapid mutual adjustments occurred as the system tended
towards an equilibrium state. The evolution towards an equilibrium morphology was characterized by the
geomorphic units, which reflected the flood magnitude. With increasing discharge, the geomorphic units
transitioned from a cone to a bank-attached bar as the depositional patterns were forced further downstream
and into the separation zone, with the bank-attached bar occupying the full extent of the separation zone.
When sediment concentration was fixed, and the discharge was adjusted, the morphology responded to the
combined channel flows downstream of the confluence. However, the morphological patterns were mainly
unaffected when the discharge was fixed and the sediment concentration was adjusted. Therefore, the
combined discharge determined the overall morphology and the development of specific geomorphic units,
and the sediment concentration controlled the morphological extent of the units. These aspects illustrate
that the morphological spatial patterns at mountain river confluences are unique and require special
attention for flood risk management.

## 6 Appendix

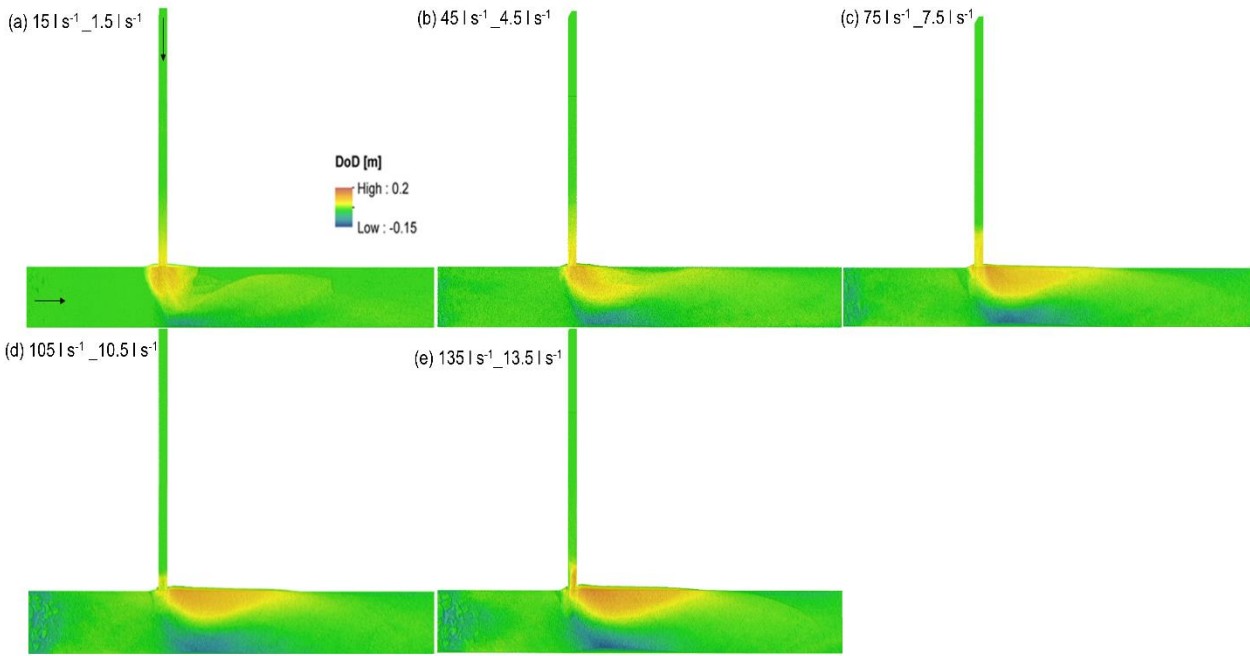

**A 1** Confluence morphology for experiments 1-5 with 5% sediment concentration, a 90° confluence angle, and a 10% tributary gradient.

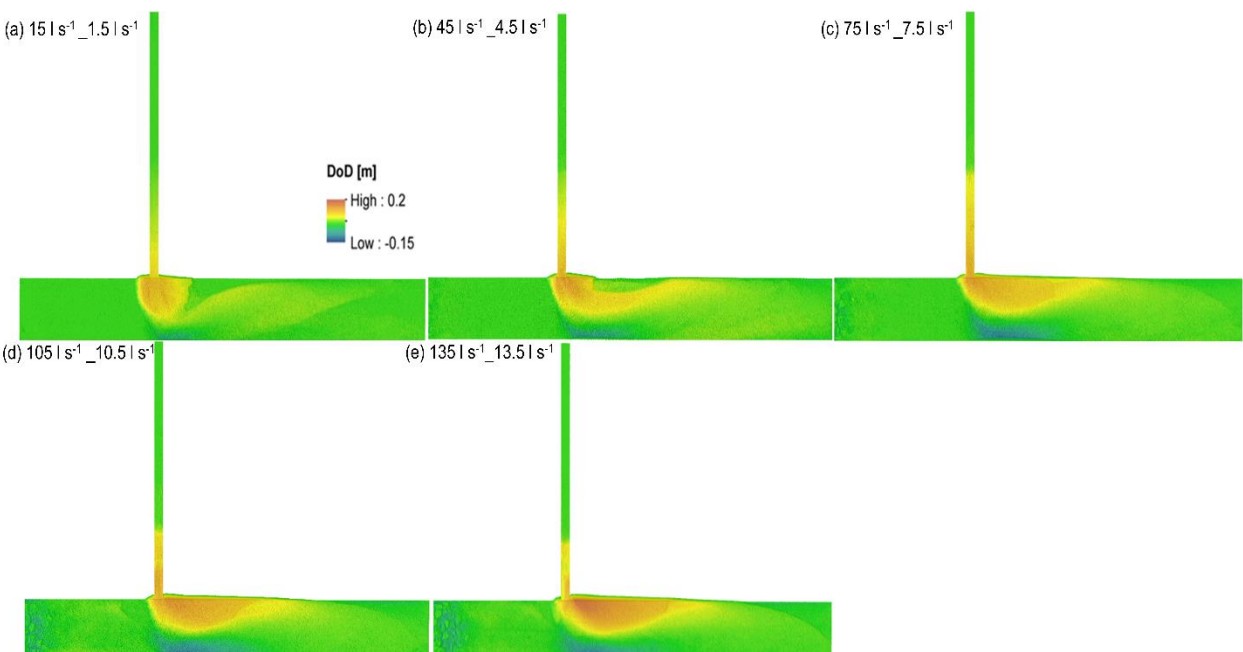

**A 2** Confluence morphology for experiments 6-10 with 7.5% sediment concentration, a 90° confluence angle, and a 10% tributary gradient.

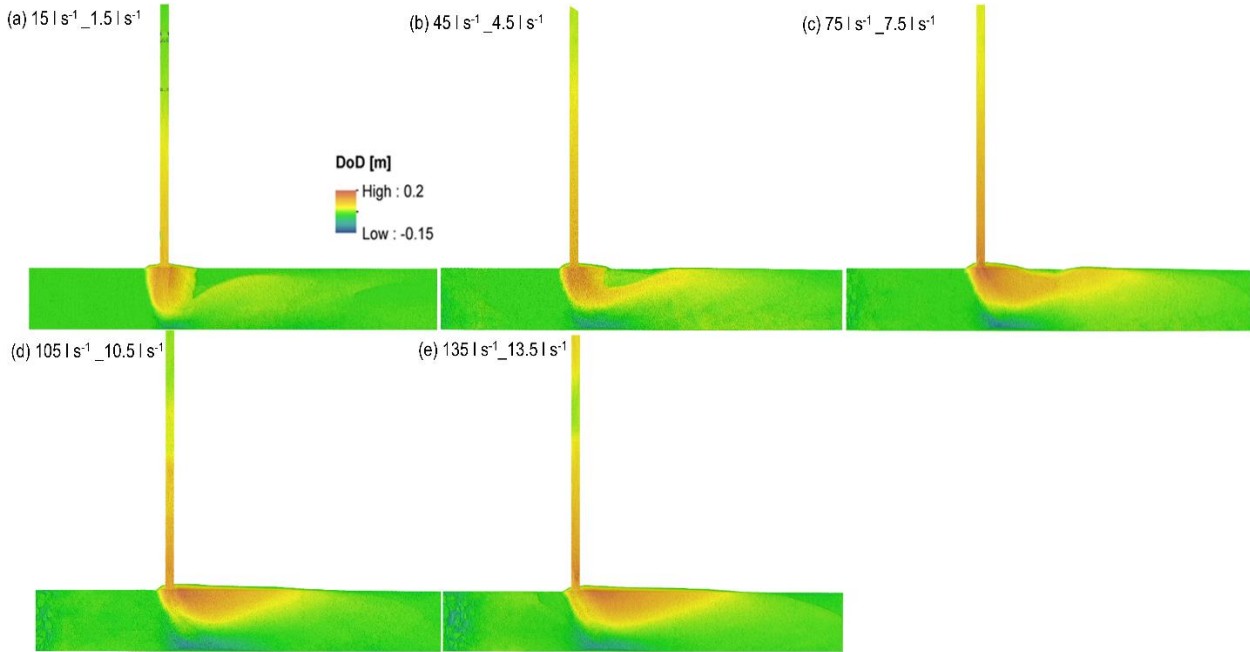

A 3 Confluence morphology for experiments 11-15 with 10% sediment concentration, a 90° confluence

angle, and a 10% tributary gradient.

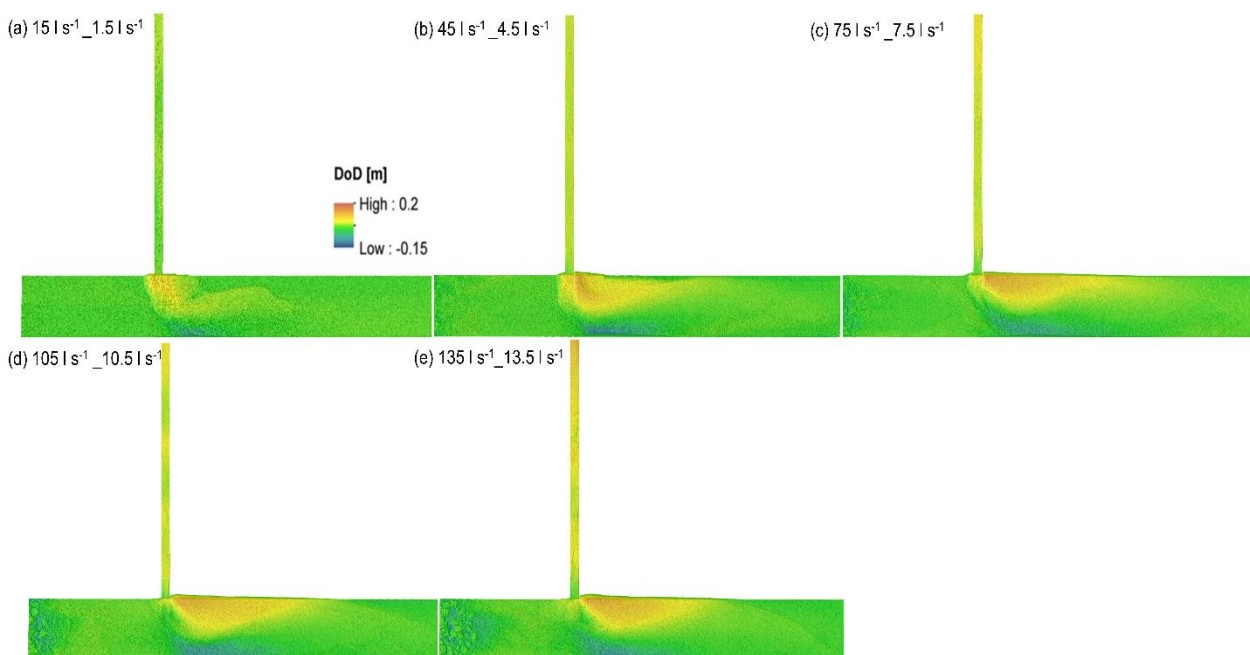

A 4 Confluence morphology for experiments 16-20 with 5% sediment concentration, a 90° confluence

angle, and a 5% tributary gradient.


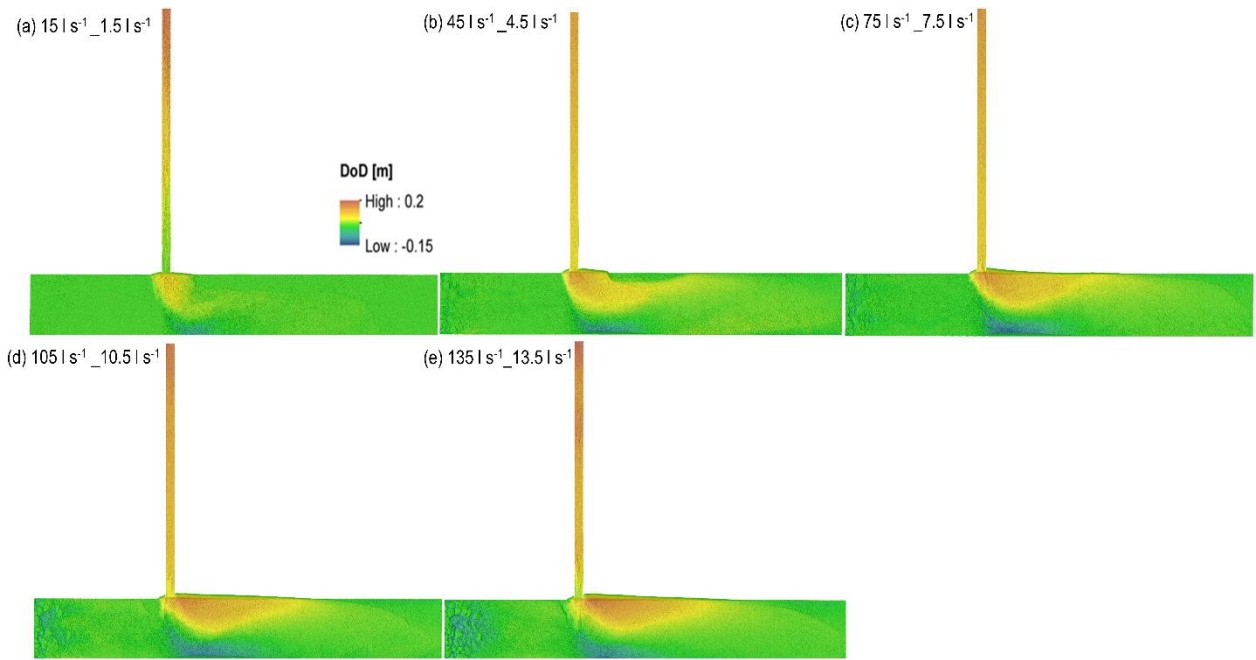


**A 5** Confluence morphology for experiments 21-25 with 7.5% sediment concentration, a 90° confluence
angle, and a 5% tributary gradient.

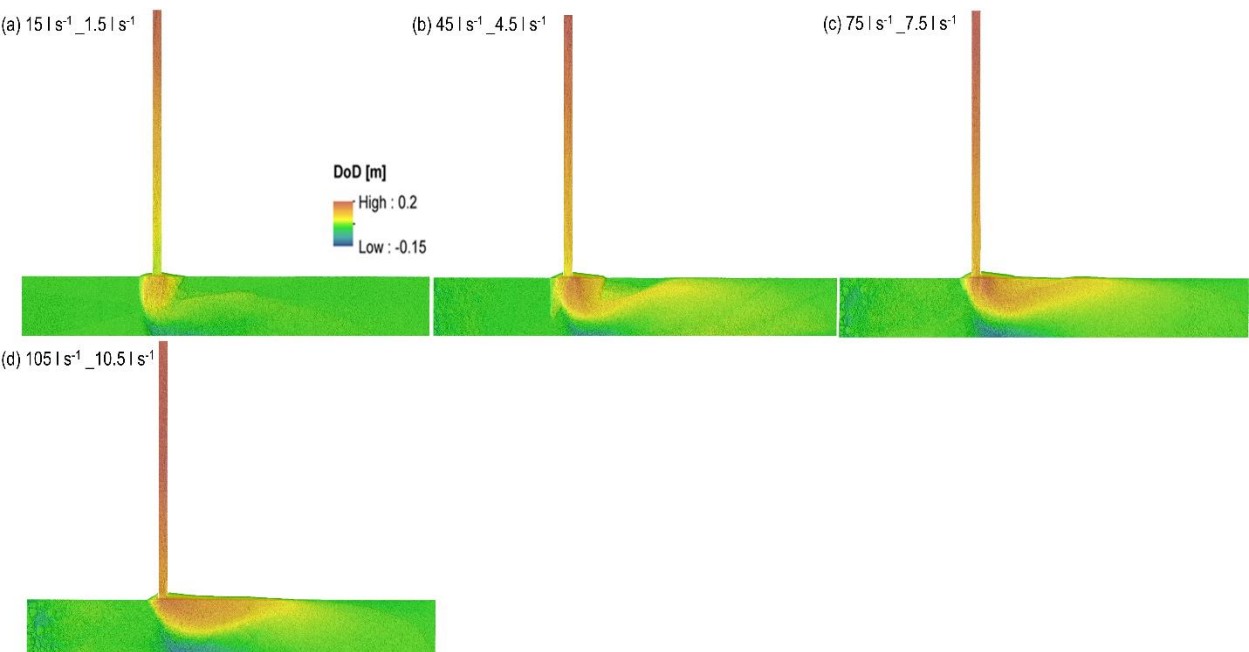


**A 6** Confluence morphology for experiments 26-29 with 10% sediment concentration, a 90° confluence
angle, and a 5% tributary gradient.

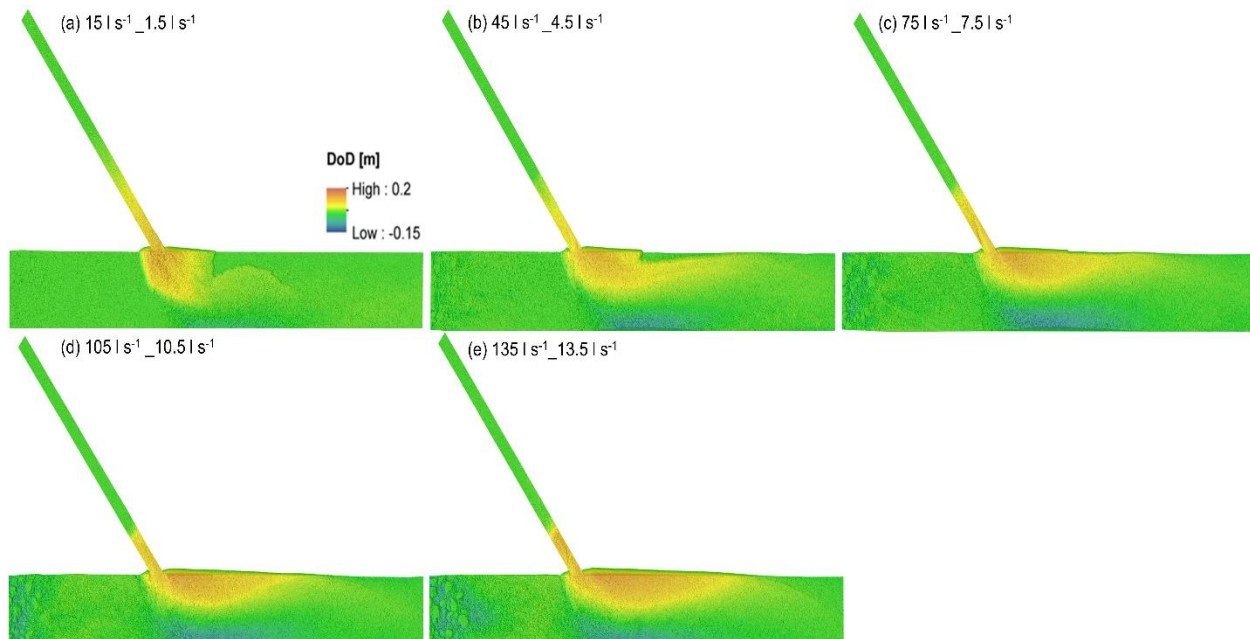

**A 7** Confluence morphology for experiments 31-35 with 5% sediment concentration, a 45° confluence

angle, and a 10% tributary gradient.

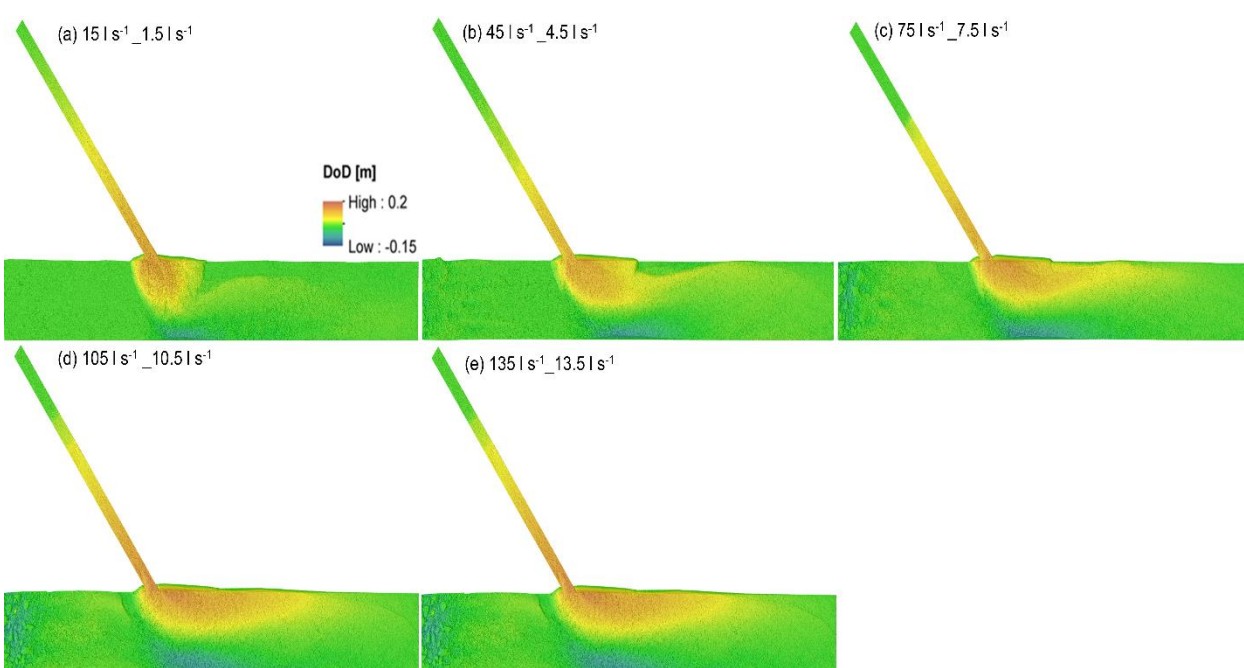

**A 8** Confluence morphology for experiments 36-40 with 7.5% sediment concentration, a 45° confluence

angle, and a 10% tributary gradient.

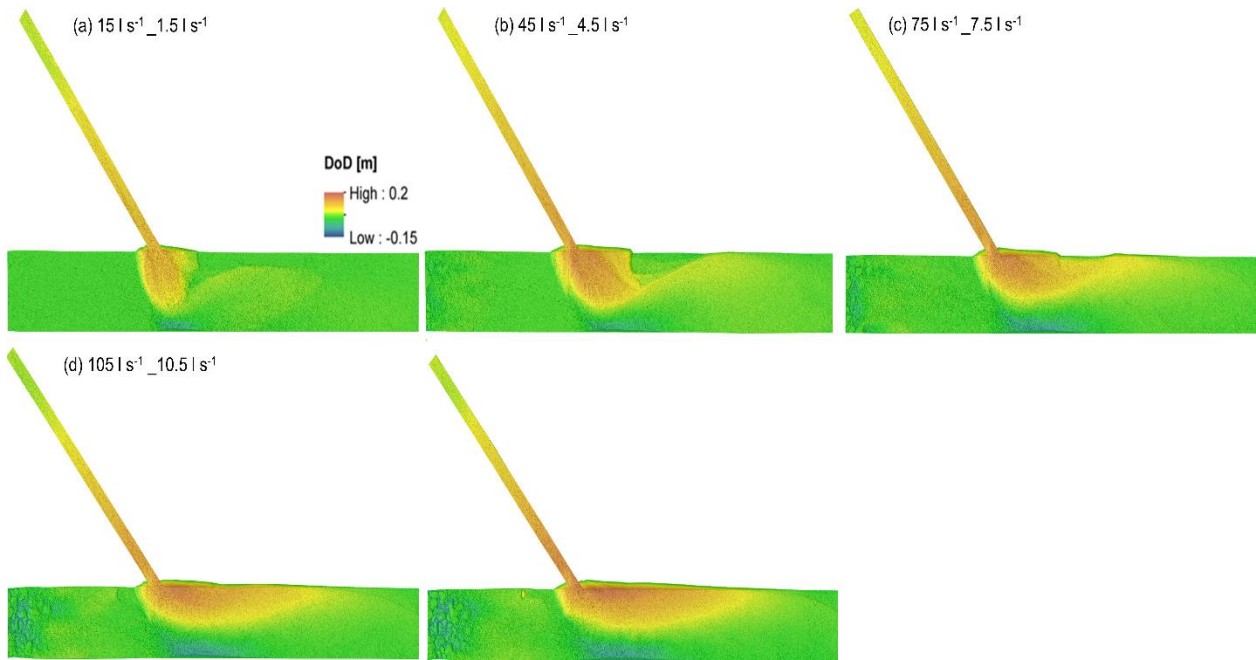


**A 9** Confluence morphology for experiments 41-45 with 10% sediment concentration, a 45° confluence
angle, and a 10% tributary gradient.
**A 10** Characteristic grain size for all experiments from samples taken in the tributary channel, the
geomorphic units (depositional or scour hole), and the recovery zone. Bold text indicates that the sampled
grain size was larger than the input mix grain size.

| Exp | D16 | | | | D50 | | | | D84 | | | | Dm | | | |
|---|---|---|---|---|---|---|---|---|---|---|---|---|---|---|---|---|
| | Trib. | Depo. | Scour | Recov. | Trib. | Depo. | Scour | Recov. | Trib. | Depo. | Scour | Recov. | Trib. | Depo. | Scour | Recov. |
| [-] | [mm] | [mm] | [mm] | [mm] | [mm] | [mm] | [mm] | [mm] | [mm] | [mm] | [mm] | [mm] | [mm] | [mm] | [mm] | [mm] |
| Input | 0.7 | 0.7 | 0.7 | 0.7 | 1.4 | 1.4 | 1.4 | 1.4 | 6.2 | 6.2 | 6.2 | 6.2 | 2.8 | 2.8 | 2.8 | 2.8 |
| 1 | 0.5 | 0.6 | 0.7 | 0.6 | 0.8 | **1.4** | **2.2** | **1.7** | 2.4 | 4.3 | **9.2** | 6.2 | 1.8 | 2.5 | **4.0** | **3.0** |
| 2 | 0.5 | 0.6 | 0.7 | 0.6 | 0.9 | **1.7** | **2.1** | **1.5** | 2.5 | 5.6 | **9.4** | **6.5** | 1.5 | **2.9** | **4.1** | **3.1** |
| 3 | 0.4 | 0.5 | 0.7 | 0.5 | 0.8 | 0.9 | **2.2** | 1.4 | 1.6 | 2.9 | **9.6** | 6.0 | 1.1 | 1.8 | **4.1** | **2.9** |
| 4 | 0.5 | 0.5 | 0.6 | 0.6 | 0.9 | 0.9 | **1.7** | 1.4 | 3.7 | 2.9 | **8.6** | **6.5** | 2.6 | 1.9 | **3.8** | **3.3** |
| 5 | 0.6 | 0.5 | 0.7 | 0.6 | 0.9 | 0.8 | **2.5** | 1.3 | 2.8 | 2.0 | **10.0** | 6.2 | 2.0 | 1.5 | **4.4** | **3.2** |
| 6 | 0.3 | 0.6 | 0.6 | 0.6 | 0.6 | **1.6** | **1.9** | **1.7** | 1.3 | 4.7 | **7.6** | **6.2** | 1.2 | **2.8** | **3.6** | **3.4** |
| 7 | 0.4 | 0.6 | **0.8** | 0.6 | 0.7 | 1.0 | **3.4** | **1.6** | 0.9 | 3.2 | **12.3** | **6.5** | 0.8 | 2.0 | **5.7** | **3.2** |
| 8 | 0.4 | 0.6 | 0.6 | 0.5 | 0.8 | 1.3 | **2.0** | 1.2 | 1.6 | 3.8 | **9.1** | 6.0 | 1.1 | 2.4 | **4.0** | **3.1** |
| 9 | 0.6 | 0.6 | **0.7** | 0.6 | 0.9 | 0.9 | **2.3** | 1.4 | 1.9 | 3.7 | **7.3** | **6.7** | 1.4 | 2.5 | **3.8** | **3.5** |
| 10 | 0.5 | 0.4 | 0.7 | 0.7 | 0.9 | 0.8 | **1.9** | **1.8** | 3.0 | 2.0 | **9.3** | 6.4 | 1.9 | 1.3 | **4.0** | 2.4 |
| 11 | 0.6 | **0.8** | **0.7** | **0.7** | 0.8 | **1.8** | **2.4** | **2.3** | 1.9 | 5.3 | **10.3** | **9.1** | 1.7 | **3.0** | **4.5** | **4.2** |
| 12 | 0.5 | 0.7 | **0.7** | **0.7** | 0.8 | **1.6** | **2.8** | **3.0** | 2.6 | 3.9 | **11.2** | **11.2** | 1.6 | 2.6 | **5.0** | **3.2** |
| 13 | 0.5 | 0.6 | **0.9** | 0.7 | 0.7 | 1.1 | **5.8** | **1.5** | 1.0 | 3.2 | **13.1** | **7.2** | 0.9 | 1.9 | **6.8** | **3.4** |
| 14 | 0.4 | 0.6 | **0.8** | 0.6 | 0.8 | 1.3 | **5.4** | 1.3 | 1.9 | 3.4 | **13.0** | 4.4 | 1.3 | 2.1 | **6.6** | 2.7 |
| 15 | 0.6 | 0.6 | **0.8** | 0.7 | 0.8 | 0.9 | **3.1** | **1.6** | 1.8 | 2.7 | **11.0** | 6.1 | 1.2 | 1.8 | **5.0** | **3.8** |
| 16 | **0.8** | 0.6 | 0.7 | 0.6 | **2.0** | **1.7** | **1.9** | **1.7** | **6.4** | 5.6 | 3.4 | **6.4** | **3.5** | **3.2** | **3.8** | **3.2** |
| 17 | 0.4 | 0.7 | **0.9** | 0.7 | 0.7 | **1.7** | **4.1** | **1.8** | 0.9 | 5.2 | 3.8 | **6.8** | 0.7 | **2.9** | **5.9** | **3.6** |
| 18 | 0.3 | 0.6 | 0.6 | 0.6 | 0.6 | 0.9 | **1.9** | **1.6** | 1.0 | 2.6 | 3.7 | 6.0 | 0.8 | 1.5 | **3.8** | **3.1** |
| 19 | 0.4 | 0.6 | **0.8** | 0.7 | 0.7 | **1.4** | **3.3** | **1.8** | 1.0 | 5.3 | 3.8 | **7.0** | 0.9 | **2.8** | **5.1** | **3.5** |
| 20 | 0.4 | 0.7 | 0.7 | 0.6 | 0.8 | **2.2** | **2.6** | 1.4 | 1.7 | **6.4** | 3.9 | **6.4** | 1.1 | **3.4** | **4.6** | **3.1** |
| 21 | **0.7** | **0.7** | **0.7** | 0.7 | **1.7** | **2.4** | **2.5** | **2.0** | **6.4** | **7.6** | 3.6 | **7.9** | **3.2** | **3.9** | **4.4** | **3.7** |
| 22 | 0.5 | **0.8** | **0.8** | 0.7 | 0.7 | **1.9** | **3.3** | **1.7** | 1.0 | 4.9 | 4.1 | **6.6** | 0.9 | **3.0** | **5.8** | **3.3** |
| 23 | 0.4 | 0.7 | **0.7** | 0.6 | 0.8 | **1.4** | **2.8** | **1.6** | 1.4 | 4.8 | 3.8 | **7.1** | 1.0 | 2.7 | **4.7** | **3.4** |
| 24 | 0.5 | 0.6 | 0.7 | 0.6 | 0.8 | 1.3 | **2.4** | **1.6** | 1.6 | 4.3 | 3.7 | 6.0 | 1.1 | 2.6 | **4.3** | **3.3** |
| 25 | 0.5 | 0.6 | **0.8** | 0.6 | 0.8 | 1.0 | **3.4** | **1.7** | 2.2 | 3.7 | 3.8 | **6.8** | 1.4 | 2.1 | **5.3** | **3.4** |
| 26 | **0.7** | **0.8** | **0.7** | 0.7 | **1.6** | **2.3** | **2.6** | **2.0** | 5.0 | **7.8** | 3.8 | **7.7** | **2.9** | **4.0** | **4.8** | **3.8** |
| 27 | 0.5 | **0.9** | **0.8** | 0.7 | 0.8 | **2.3** | **3.1** | **1.7** | 1.0 | 5.6 | 3.8 | **6.8** | 0.9 | **3.3** | **5.1** | **3.4** |
| 28 | 0.5 | **0.7** | **0.8** | 0.7 | 0.8 | **1.6** | **3.1** | **1.7** | 1.9 | 3.7 | 3.9 | **7.8** | 1.5 | 2.4 | **5.4** | **3.7** |
| 29 | 0.5 | 0.7 | **0.7** | 0.7 | 0.8 | **1.7** | **2.6** | **1.8** | 1.8 | 5.9 | 3.8 | **6.6** | 1.3 | **3.2** | **4.6** | **3.4** |
| 30 | - | - | - | - | - | - | - | - | - | - | - | - | - | - | - | - |
| 31 | 0.6 | **0.7** | **0.8** | 0.7 | 0.9 | 1.9 | **2.8** | **1.9** | 2.6 | 5.7 | **10.0** | **6.9** | 2.0 | **3.2** | **4.6** | **3.6** |
| 32 | 0.5 | 0.5 | 0.7 | 0.7 | 0.8 | 0.9 | **2.0** | **1.9** | 1.7 | 3.5 | **7.1** | **7.5** | 1.2 | 2.1 | **3.7** | **3.7** |
| 33 | 0.6 | 0.52 | 0.7 | 0.7 | 0.9 | 0.8 | **1.5** | **1.7** | 2.8 | 2.2 | 6.0 | **7.6** | 1.9 | 1.5 | **3.1** | **3.6** |
| 34 | 0.6 | 0.6 | **0.7** | 0.7 | 0.9 | 0.9 | **1.9** | **1.8** | 2.9 | 3.5 | **6.4** | **7.3** | 1.8 | 2.1 | **3.3** | **3.5** |
| 35 | 0.6 | 0.5 | **0.8** | 0.7 | 1.2 | 0.8 | **3.0** | **1.7** | 3.8 | 1.7 | **11.0** | **6.6** | 2.6 | 1.3 | **5.1** | **3.4** |
| 36 | 0.6 | **0.8** | **0.7** | 0.7 | 0.9 | **2.2** | **2.7** | **1.8** | 2.8 | 5.9 | **10.1** | **7.1** | 2.1 | **3.3** | **4.6** | **3.7** |
| 37 | 0.5 | 0.7 | 0.7 | 0.7 | 0.8 | **1.5** | **1.6** | **1.7** | 1.7 | 5.9 | **9.0** | **7.2** | 1.2 | **3.1** | **3.8** | **3.7** |
| 38 | 0.5 | 0.6 | **0.7** | 0.6 | 0.8 | 0.9 | **2.2** | **1.5** | 1.6 | 4.0 | **8.6** | 5.8 | 1.1 | 2.4 | **4.0** | **3.1** |
| 39 | 0.6 | 0.55 | **0.7** | 0.6 | 0.8 | 0.9 | **2.5** | 1.4 | 1.9 | 3.4 | **9.8** | 5.7 | 1.3 | 2.1 | **4.4** | **3.0** |
| 40 | 0.6 | 0.5 | **0.7** | 0.7 | 1.0 | 0.8 | **2.5** | **1.9** | 3.3 | 1.9 | **10.0** | 5.9 | 2.0 | 1.3 | **4.4** | **3.3** |
| 41 | **0.7** | **0.9** | 0.7 | 0.7 | **1.7** | **3.4** | **1.8** | **1.9** | 4.0 | **8.5** | 7.3 | **7.8** | **2.9** | **4.7** | 3.5 | **3.8** |
| 42 | 0.5 | **0.9** | 0.7 | **0.7** | 0.8 | **2.3** | **2.5** | **1.9** | 1.0 | 4.8 | **10.4** | **6.3** | 0.9 | **3.1** | **4.6** | **3.3** |
| 43 | 0.5 | 0.6 | **0.7** | 0.7 | 0.8 | 1.1 | **2.4** | **1.8** | 1.1 | 3.6 | **9.4** | **7.8** | 1.0 | 2.2 | **4.3** | **3.8** |
| 44 | 0.6 | 0.6 | **0.7** | **0.7** | 0.9 | 1.0 | **2.3** | **2.0** | 1.8 | 3.0 | **9.7** | **7.7** | 1.3 | 1.9 | **4.3** | **3.7** |
| 45 | 0.6 | **1.2** | **0.8** | **0.8** | 0.8 | **1.9** | **2.5** | **2.5** | 1.8 | 5.0 | **10.4** | **7.4** | 1.4 | **3.02** | **4.6** | **3.9** |

**7    Data Availability**

Data are available from the corresponding author upon reasonable request.

**8    Author Contributions**

TSPO: Conceptualization, data curation, formal analysis, investigation, methodology, visualization, writing
– original draft preparation (with input from all co-authors). TK: Formal analysis, data curation. BM:
Conceptualization, methodology, writing – review and editing. JH: Formal analysis, investigation, writing –
review and editing. AA: Conceptualization, writing – review and editing. FC: Conceptualization, supervision,
project administration, funding acquisition, writing – review and editing BG: Conceptualization, supervision,
project administration, funding acquisition, writing – review and editing.

**9    Competing Interests**

The authors declare that they have no conflict of interest.

**10    Acknowledgements**

The authors would like to thank the Autonomous Province of Bolzano - South Tyrol – Department of
Innovation, Research, University and Museums for funding the project: Towards an Efficient Design of River
Confluences to Manage Intense Sediment Impacts from Tributary Torrents (ECOSED_TT, contract number
24/34). This project and the accompanying funding provided the framework to conduct detailed
investigations into mountain-river confluence hydraulics and morphology. Additionally, we acknowledge the
funding of the Project Anid/Conicyt Fondecyt Regular Folio 1200091 titled "Unravelling the Dynamics and
Impacts of Sediment-Laden Flows in Urban Areas in Southern Chile as a Basis for Innovative Adaptation
(sedimpact)" led by the PI Bruno Mazzorana.

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
