# Peer review of "Limited effect of the confluence angle and tributary gradient on Alpine confluence"

_EGUsphere, 2023_

## Author Comment (AC1)

Response to the reviewer:

We are thankful and appreciative for the constructive comments and feedback that have helped us improve the overall quality and clarity of the manuscript. Specifically, dealing with the reasons behind the different morphological responses of mountain river confluences vs. lowland confluences. We have included a response to the reviewers' main comments/conclusions below, the line-by-line comments are addressed here, have been implemented, and will be reflected in the re-submitted manuscript one the discussion period has ended. Our response is the blue text.

**Summary**

The manuscript presents findings from laboratory studies on confluences in mountain streams. It builds upon a recently published work by St. Pierre Ostrander et al. (2023). In this paper, the experimental program has been extended by 30 additional tests to focus on the effect of the tributary gradient and confluence angle. The experiments were conducted for different total discharges, sediment concentration, tributary gradient, and confluence angle. The model values were based on information of more than 100 confluences in Italy and Austria. The results demonstrate that the discharge and sediment concentration affect the morphology of the confluence. In contrast, tributary gradient and confluence angle had only a minor effect. For all tests, similar morphological features have been observed, including deposition cone, separation zone bar, or scour.

**Main comments / Conclusion**

The paper is well written and the differences or extensions compare to St. Pierre Ostrander et al. (2023) are clear. The experiments have been conducted thoroughly and the findings are presented in a comprehensive way. I have the following main comments:

1. Main outcome of the paper is that the confluence angle does not affect confluence morphology, which is different compared to low-land rivers. Please add to Introduction and Conclusions what the findings of the low-land rivers literature were with respect to the confluence angle and describe the physics behind it. Why is this different for mountain streams?

Response: We agree with the reviewer's comment that detail was lacking regarding the confluence angle, we have added details which has improved the clarity of the manuscript. In lowland regions increasing both the discharge ratio and the confluence angle leads to a greater mutual deflection of flows and a larger separation zone which is the largest sink for tributary transported sediment (Best, 1987). Flow deflection influences the shear layers generated between the two converging flows along which powerful vortices are created and are responsible for increased bed shear stresses (Mosley, 1976; Best, 1987; Penna et al., 2018; De Serres et al., 1999). Decreasing the confluence angle results in a greater mixing of flows, a smaller separation zone, and declined levels of turbulence in the junction (Best, 1988; Penna et al., 2018). The most apparent differences result from the elevated intensity of hydraulic and hydrologic processes occurring in steep mountain channels (Rickenmann, 2016) relative to the much less intense conditions found in lowland channels. This is apparent considering the smaller Froude numbers and velocities measured in studies dealing with lowland confluences (e.g., Best, 1988; Biron et al., 1996), compared to the Froude numbers and velocities from our study and velocities from steep tributaries in the study region (e.g., Hübl et al., 2005). The higher Froude numbers and associated velocities not only intensify the event (Rudolf-Miklau et al., 2013) and the rate of channel adjustments (Wohl, 2010) but can also support equal mobility sediment transport conditions (Rickenmann, 2016) which can deliver massive amounts of sediment to the confluence, creating morphological feedback where the morphology reacts to the intensity of the event to maximize sediment transport capacity (White et al., 1982; Wohl,

2010). This explains why we observed the same general morphological patterns for different geometric configurations, given the same hydraulic parameters and sediment supply rates. Additional text and a table of hydraulic variables from the study have been added to the introduction (L69-85) and to the conclusion (L557-578) which was rewritten to better convey the key points as suggested by the reviewer.

Best, J. L.: Sediment transport and bed morphology at river channel confluences, Sedimentology, 35, 481-498, doi:10.1111/j.1365-3091.1988.tb00999.x, 1988.

Biron, P., Roy, A. G., & Best, J. L.: Turbulent flow structure at Concordant and discordant open-channel confluences. Exp Fluids, 21(6), 437–446, doi:10.1007/bf00189046, 1996.

De Serres, B., Roy, A., Biron, P., and Best, J. L.: Three-dimensional flow structure at a river channel confluence with discordant beds, Geomorphology, 26, 313-335, doi:10.1016/S0169-555X(98)00064-6, 1999.

Hübl, J., Ganahl, E., Bacher, M., Chiari, M., Holub, M., Kaitna, R., Prokop, A., Dunwoody, G., Forster, A., Schneiderbauer, S.: Dokumentation der Wildbachereignisse vom 22./23. August 2005 in Tirol, Band 1: Generelle Aufnahme (5W-Standard); IAN Report 109 Band 1, Institut für Alpine Naturgefahren, Universität für Bodenkultur-Wien (unveröffentlicht), 2005.

Mosley, M. P.: An experimental study of channel confluences, J. Geol., 84(5), 535-562, doi:10.1086/628230, 1976

Penna, N.; De Marchis, M.: Canelas, O.B.; Napoli, E.; Cardoso, A.H.; Gaudio, R.: Effect of the Junction Angle on Turbulent Flow at a Hydraulic Confluence. Water, 10, 469, doi:10.3390/w10040469, 2018.

Rickenmann, D. (Ed.).: Methods for the Quantitative Assessment of Channel Processes in Torrents (Steep Streams). CRC Press, 2016.

Rudolf-Miklau, F., Suda, J. Design Criteria for Torrential Barriers. In: Schneuwly-Bollschweiler, M., Stoffel, M., Rudolf-Miklau, F. (eds) Dating Torrential Processes on Fans and Cones. Adv. Glob. Change Res., vol 47. Springer, Dordrecht, doi:10.1007/978-94-007-4336-6_26, 2013.

White, W. R., R. Bettess, and E. Paris: Analytical approach to river regime, J. Hydraul. Div. Am. Soc. Civ. Eng., 108, 1179– 1193, doi:10.1061/JYCEAJ.0005914, 1982.

Wohl, E. E..: Mountain rivers revisited. American Geophysical Union/Geopress ISBN 9780875903231, 2010.

2. The focus of this paper was on confluence angle and tributary gradient: why did the paper include only 2 different angles and 2 different gradients, while 5 total discharges and 3 different sediment concentrations (per discharge) were tested? In addition, why did you decide to not vary the discharge ratio, as this was also mentioned in the outlook of St. Pierre Ostrander et al. (2023)?

Response: The choice of geometric adjustments was based on the most occurring values in the study region for the confluence angle and tributary gradient. The reason why our study is limited to the presented channel configurations is the time, financial commitment, and project duration required to make additional changes. The discharge ratio was fixed, although adjustments were mentioned in the outlook of St. Pierre Ostrander et al. (2023), so that we could have fully comparable sets of experiments where any morphological response is from the changes in hydraulics and sediment transport from geometric changes. If limiting factors were not an issue, an ideal experiment plan would have the same 45 experiments run again, but with a different discharge ratio. However, Holzner et al. (2024) used the first 15 experiments from this study to validate a 2D numerical model (BASEMENT). With the validated model they test the effects of doubling the discharge ratio using an upscaled (30) version of the physical model used in this study.

The discharges represent flood conditions and one extreme event in the study region, and varying sediment supply rates have been introduced in some of the first studies dealing with confluence morphology (Mosley, 1976). Testing these discharges allowed us to determine if certain morphologies correspond to certain RI events, while the 3 sediment concentration groups enabled an assessment of morphologies as they relate to different sediment concentration events where varying levels of transport capacity utilization may affect confluence morphodynamics.

Holzner, J., Ostrander, T. St., Andreoli, A., Mazzorana, B., Comiti, F., & Gems, B.: 2D numerical modeling of intense bedload-transport processes at confluences of Mountain Rivers and steep tributaries. Nat Hazards. doi:10.1007/s11069-023-06212-6, 2024.

Mosley, M. P.: An experimental study of channel confluences, J. Geol., 84(5), 535-562, doi:10.1086/628230, 1976

St. Pierre Ostrander, T., Holzner, J., Mazzorana, B., Gorfer, M., Andreoli, A., Comiti, F., and Gems, B.: Confluence morphodynamics in Mountain Rivers in response to intense tributary bedload input, Earth Surf. Processes, 1-22, doi:10.1002/esp.5613, 2023.

3. How was "equilibrium" defined? Was it achieved after the 20 minutes of test run and did it not take longer? See for example Ancey (2020a,b): https://doi.org/10.1080/00221686.2019.1702594 and

https://www.tandfonline.com/doi/full/10.1080/00221686.2019.1702595

Response: The 2 provided references (which have been added to the text L105-108) certainly show the complications and factors which make predicting bedload transport difficult and the multitude of approaches (Ancey, 2020a), developed over the last decades to evaluate and predict bedload transport and the resulting bedforms. In this context, we understand the need to discuss how tending towards equilibrium morphology was defined. Ancey (2020b) asks a similar question in defining bed equilibrium, where researchers have stated equilibrium is met when fluctuations are slow. However, Ancey (2020b) points out that the problem of fluctuating bed load transport is rarely mentioned, which as Ancey (2020b) suggests, could be at the very core of understanding and developing more accurate approaches to predicting bed load transport. In this regard, we ensured that we described the resulting morphology as one that is tending towards an equilibrium state that is representative of the driving impact factors which is optimized to deal with the intense nature of the event.

The tending towards equilibrium morphology was defined by the stable, large-scale, reoccurring, geomorphic units which reoccurred for every experiment with identical input

conditions. From video registrations and direct observations, we were able to establish that the final morphology is reached after a relatively short duration and remains stable throughout the experiment. This morphology is event-specific and can certainly be altered by changes in event characteristics, for example, discussed in Ancey (2020a) the mixing of fast and slow processes (amongst others). Additionally, Ancey (2020b) discusses how rivers are closer to punctuated equilibrium systems, with rapid changes in bed morphology over short time intervals, then followed by long periods of stasis. The tending towards equilibrium morphology discussed in the presented work represents these punctuated moments of intensity, that would certainly be re-worked during prolonged periods of weak activity. Being that the presented work deals exclusively with torrential hazard events the speed of the processes was consistently fast (Wohl, 2010) as such the large-scale morphology, adapted to the intensity of the event remained stable once established.

Furthermore, Holzner et al. (2024) used the results from the physical model, used in this study (EXP 1-15), to validate a 2D morphological numerical model (BASEMENT). To ensure that the resulting morphology did indeed represent a system tending towards morphological equilibrium the 2D numerical model was run for a duration of 54h, which corresponds to a 10h experimental run, which is far longer than a typical torrential hazard event. The results confirmed that the morphology captured at the end of a physical experiment was indeed a stable system tending towards equilibrium. The reason for morphological equilibrium to be established in a short relative time frame deals with the intensity of modelling torrential events. Processes occur much faster because of the gradient of the tributary channel and high sediment loads and discharges (See response to comment 1).

Ancey, C.: Bedload transport: A walk between randomness and determinism. part 1. the state of the art. J. Hydraul. Res., 58(1), 1–17. doi:10.1080/00221686.2019.1702594, 2020a

Ancey, C.: Bedload transport: A walk between randomness and determinism. part 2. challenges and prospects. J. Hydraul. Res., *58*(1), 18–33. doi:10.1080/00221686.2019.1702595, 2020b

Wohl, E. E..: Mountain rivers revisited. American Geophysical Union/Geopress, Washington, D.C., ISBN 9780875903231, 2010.

Holzner, J., Ostrander, T. St., Andreoli, A., Mazzorana, B., Comiti, F., & Gems, B.: 2D numerical modeling of intense bedload-transport processes at confluences of Mountain Rivers and steep tributaries. Nat Hazards. doi:10.1007/s11069-023-06212-6, 2024.

I provide additional comments per line below. Based on my review, I recommend minor revision in form and content.

**Comments per section/line**

Response: All revisions that have been suggested by the reviewer have been implemented. Line numbers reflect the current version of the revised manuscript but will certainly change as we continue to revise based on further comments and suggestions.

1. L116: Please add why these hypotheses have been formulated; what potential processes or governing parameters lead to these hypotheses?

   Response: Details into the formulation of the hypotheses have been added (L141-146) and primarily deal with the conclusions drawn from Lane (1955), St. Pierre Ostrander et al. (2023), and White et al. (1982). St Pierre Ostrander et al. (2023) established additional factors besides the confluence angle and discharge ratio influence confluence morphology (hypothesis 1), and Lane (1955) and White et al. (1982) describe how a channel will react to the impact conditions to establish an event based morphological equilibrium (hypothesis 2) to maximize or minimize the value of a specific parameter, like sediment transport, for example.

2. L151: see also main comment 2, but why 5 discharges and why steady-state?

Response: The 5 discharges represent flood conditions and one extreme event. This allowed us to determine if morphological patterns correspond to certain reoccurrence intervals while also thoroughly representing the hydraulic conditions in the study region. Steady-state modelling was used so that morphological development could more easily be associated with one of the introduced factors and to make the morphological development comparable to research dealing with lowland confluences, which largely assume steady-state conditions. An unsteady hydrograph would make it difficult to discern at what point (rising limb etc.) the morphology reacts to the impact factors and to which one. Steady-state modelling mitigates this uncertainty and in this application is consistent with other modelling approaches regarding the hydrodynamics and morphological development of mountain river confluences (e.g., Roca et al., 2009; Leite Ribeiro et al., 2012). Text has been added to clarify this component of the experimental plan (L177-184).

3. L155ff: Please add the accuracy of the measurement devices.

Response: The accuracy of the measurement devices has been added to the text following the reviewer's suggestions (L187-189).

4. L162: Why 20 minutes? Which scaling factor did you choose and what was the reference value to derive this duration?

Response: Scaling was done according to Froude similarity; transferring model dimensions to nature allows a scale factor range of 20-40. The scale is determined by the width of the tributary at the confluence relative to the width of the tributary in the physical model and was referred to as the specific Event duration was scaled down to laboratory

conditions by a factor of 30. Thirty was chosen as it is the median scale factor the physical model is designed to accommodate. The event duration is based on incident reports of torrential events occurring within the study region compiled by the Tyrol Torrent and Avalanche Control Agency (WLV) and other sources of event documentation, for example, Hübl et al. (2012). Text has been added to clarify this aspect L196-197.

5. Table 1: Not all parameters have been introduced in the text or in the table (e.g., Qm, Qt). Please check entire manuscript. Please add Froude number and stream power to the table.

Response: Variable descriptions have been added to the table caption (L212-215). Unit stream power for all experiments is summarized in Table 4, and clear water hydraulic variables are added to Table 1 (L82). The main channel was modified with a fixed bed for all geometric configurations to obtain these values since direct flow field measurements were not possible during an experiment. The values in Table 1 represent the undisturbed, initial conditions at the onset of an experiment and are indicators of the initial hydraulic conditions which initiate the depositional or erosional patterns in the confluence.

6. Equation 1: please introduce right after you first mention it and also introduce the parameter.

Response: The equation has been moved to immediately follow its introduction (L256).

7. L244: "Discharges and related unit stream power above 45 l s-1" … please add unit stream power or write 45 l/s after discharge, otherwise confusing.

Response: The text has been revised to reflect the reviewers' comment (L281).

8. L249: Reference to subcritical flows – would be interesting to know Froude numbers of this study (see recommendation for Table 1)

Response: Froude numbers have been added to Table 1.

9. Section 3.2+3.3.: Consider stating the value of the tributary gradient in addition to the test number; difficult to remember the test number that refers to a certain gradient. This would be especially helpful in the Figure captions and I recommend making them consistent.

Response: Geometrical parameters and experiment numbers have been added to the text in both sections. We agree with the reviewer that additional details were required to impart greater clarity (L317-318, L320, L325, L330-331, L334, L337-338, L340-341).

10. L278: Here, I recommend to state the gradient value instead of "from the decrease in gradient". For example: "A smaller tributary gradient of 5% led to reduced velocity and … compared to the depositional forms with a tributary gradient of 10%" or similar.

Response: The suggestion has been added to the text as we agree that clarification is needed when discussing the sets of experiments (L315-318).

11. Figure 4: figure caption "supporting a qualitative representation of morphological differences" is different to L279 that the gradient did not have an effect. I recommend deleting the sentence on the qualitative representation in the figure caption.

Response: The sentence has been deleted according to suggestion (L325).

12. L295: add gradients and add test numbers in brackets + refer to Table 1

Response: Geometrical parameters and experiment numbers have been added in brackets and hydraulic variables have been compiled in Table 1 which has been referenced in the text (L329-337).

13. Figure 5: please add test number.

Response: Experiment numbers have been added to impart greater clarity as suggested (L341-342).

14. Figure 6: please add tributary gradient

Response: The tributary gradients have been added as suggested (L358-359).

15. L329: I recommend deleting the sentence on the qualitative representation in the figure caption.

Response: The sentence has been deleted (L370-371).

16. Figure 8: please add test number.

Response: The experiment numbers have been added to the caption to further clarify the figure (L382-384).

17. L349: Where are the results of the depositional volume plotted? Please add.

Response: A reference to the figure has been added to the text (L391).

18. L356: Please revise this sentence; the statement is not clear to me. Consider adding the values of the confluence angle for clarification instead of "greater" – to what?

Response: The value for the confluence angle has been added in place of "greater" (L397).

19. Figure 9: add angle to the caption

Response: The corresponding gradients have been added to the caption to add clarity (L400-401).

20. Table 5+6: add sigma also to the last 3 columns

Response: The sigma denotes the standard deviation, and the last three columns are the results of the pairwise post hoc mean comparison testing which uses letters to represent differences in means. The table caption required clarification to convey this and has been revised (L424-425, L456-458).

21. Figure 10: add reference to Table 4

Response: A reference has been added to further clarify the figure (L438) and has also been added to the following boxplot figures to maintain consistency (Figure 11, L467 and Figure 12, L489).

22. L434: Add more details on why turbulence increasing with increasing confluence angle.

Response: Additional details have been added to the text explaining the mechanisms behind the increased turbulence in the junction (L474-476).

23. Table 7: Why was a different statistical test used compared to the other factors? T-Test for angle compared to ANOVA for discharge and sediment concentration?

Response: The confluence angle has 2 groups, 45° and 90°, the sediment concentration 3 groups, 5%, 7.5% and 10%, while the combined discharge has 5 groups, 16.5 l/s, 49.5 l/s, 82.5 l/s, 115.5 l/s, and 148.5 l/s The number of groups (and their distribution, and variance) determined the applied tests (Figure 2). A T-test requires 2 groups while ANOVA

testing requires 3 groups. Text has been added to the caption of Fig. 2 to add clarity to this component (L246-247).

Sawyer, S. F.: Analysis of variance: The Fundamental Concepts. Journal of Manual & Manipulative Therapy, 17(2). doi:10.1179/jmt.2009.17.2.27e, 2009b

Witte, R. S., & Witte, J. S.: Statistics 11th Edition. Wiley and Sons, Hoboken, New Jersey ISBN 1119386055, 2017.

24. L452: See main comment 1. Please add references and briefly summarize physics behind it so the reader understands the differences between lowland and mountain streams.

Response: Additional details and references (below) have been added to the discussion section to further describe the factors and conditions that influence mountain river confluences and how they differ from lowland conditions (L494-508). We discuss how the confluence adjusts to the intensity of flooding and associated bed load transport occurring in steep channels and how this intensity does not occur in lowland channels, causing a different morphological response.

Darby, S. E., and M. J. Van De Wiel: Models in fluvial geomorphology, in Tools in Fluvial Geomorphology, edited by G. M. Kondolf and H. Piégay, pp. 503–537, John Wiley, Chichester, U.K., doi:10.1002/9781118648551.ch17, 2003.

Davies, T. R. H., and A. J. Sutherland: Extremal hypotheses for river behaviour, Water Resour. Res., 19, 141– 148, doi:10.1029/WR019i001p00141, 1983.

Liu, T., Fan, B., and Lu, J.: Sediment-flow interactions at channel confluences: A flume study, Adv. Mech. Eng., 7(6), 1-9, doi:10.1177/1687814015590525, 2015.

White, W. R., R. Bettess, and E. Paris: Analytical approach to river regime, J. Hydraul. Div. Am. Soc. Civ. Eng., 108, 1179– 1193, doi:10.1061/JYCEAJ.0005914, 1982.

Yang, C. T.: Minimum unit stream power and fluvial hydraulics, J. Hydraul. Div. Am. Soc. Civ. Eng., 102, 919– 934, doi:10.1061/JYCEAJ.0004589, 1976.

Yang, C. T., and C. C. S. Song: Theory of minimum rate of energy dissipation, J. Hydraul. Div. Am. Soc. Civ. Eng., 105, 769–784, doi:10.1029/WR017i004p01014, 1979.

25. L453: I recommend to remind the reader of hypothesis 1 and explain why it was confirmed. References to main plots in paper would be helpful.

Response: Hypothesis 1 has been reiterated and figure references to plots and tables have been added to the text (L494-495).

26. L456: sympathetically? Not clear

Response: Sympathetically has been removed to impart greater clarity (L511).

27. L472: I recommend to remind the reader of hypothesis 2 and again explain why the same geomorphic units occurred.

Response: The hypothesis has been restated and an explanation for the reoccurrence of geomorphic units has been added to the text (L526-532).

28. Conclusions: see main comment 1 and consider rewriting the conclusion to add an explanation why the angle does not have an effect in mountain streams compared to lowland rivers. Last sentence comes a bit as a surprise and not so clear what is meant by sediment buffer zones.

Response: We agree with the reviewer that the conclusion required further revisions. Accordingly, the conclusion has been re-written (L565-586) with details as to why confluence angle effects are limited in the presented work.

---

## Author Comment (AC2)

Response to the reviewer:

The authors are very appreciative of the constructive feedback and thoughtful review of the presented manuscript. The reviewer brought up many good points and writing suggestions that improved the overall quality and clarity of the manuscript. Specifically, issues and potential limitations dealing with scale effects between model and prototype. We have given each comment our full attention and have implemented every suggestion, the line-by-line comments are addressed here and will be reflected in the re-submitted manuscript. Our response is in the blue text.

The authors perform a series of experiments in a scaled physical model, to better understand the geomorphological dynamics at confluences of steep, high-sediment-load tributaries, into larger rivers, during a flash flood event (in the tributary). I agree that this is an area needing more research, and opine that this work is a good contribution.

My comments are separated into major issues, minor issues, and writing suggestions/corrections.

**1. Major comments:**

A.1. The manuscript needs a more extended and improved discussion of a series of scaling issues, including:

(i) Experiment run times vs. the expected range of actual flood durations in the prototype tributaries

Response: Experimental event duration was based on natural (prototype) event duration which was established through documentation supplied by the Austrian Service for Torrent and Avalanche Control and detailed event documentation (e.g., Hübl et al., 2005; Hübl et al., 2012). From these event duration values, we (according to Froude similarity) scaled down by a factor of 30 to flume conditions. Thirty was

chosen because it is the median scale factor the physical model was designed to accommodate. Transferring model dimensions to natural conditions allows a scale factor range of 20-40. To ensure that scale issues concerning experiment run times were minimal in this regard, Holzner et al. (2024) used the physical model results to validate a 2D numerical model (30 scale) (BASEMENT), which was then run for 54h (10h model scale) to ensure that the scaled event duration was sufficient to ensure that no further major morphological development occurred and the resulting morphology was representative of the event conditions. We agree that detail was lacking in regards to experiment run times and text has been added to the manuscript to add clarity to this component (L202-L203).

Holzner, J., Ostrander, T. St., Andreoli, A., Mazzorana, B., Comiti, F., & Gems, B.: 2D numerical modeling of intense bedload-transport processes at confluences of Mountain Rivers and steep tributaries. Nat Hazards. doi:10.1007/s11069-023-06212-6, 2024.

Hübl, J., Ganahl, E., Bacher, M., Chiari, M., Holub, M., Kaitna, R., Prokop, A., Dunwoody, G., Forster, A., Schneiderbauer, S.: Dokumentation der Wildbachereignisse vom 22./23. August 2005 in Tirol, Band 1: Generelle Aufnahme (5W-Standard); IAN Report 109 Band 1, Institut für Alpine Naturgefahren, Universität für Bodenkultur-Wien (unveröffentlicht), 2005

Hübl J., Eisl J., Tadler R.: Ereignisdokumentation 2012, Jahresrückblick der Ereignisse. IAN Report 150, Band 3, Institut für Alpine Naturgefahren, Universität für Bodenkultur - Wien (unveröffentlicht), 2012

(ii) Sediment density and size, and a better explanation of how these were chosen

Response: The input mix consisted of various sizes of quartz sand grains ($\rho$ = 2650 kg m$^{-3}$) with the experimental GSD curve being based on a total of 65 volume (subsurface) and line (surface) sediment samples taken from confluences in the study region (St. Pierre Ostrander et al., 2023). These dimensions were then scaled by a factor of 30 to flume conditions. The factor of 30 was chosen for the same reasons discussed in A1 (i). Text has been added L156-165 to clarify this component of the experimental program. Quartz is typically used in gravel-bed river modelling (Young & Warburton, 1996) as the sediment density is equal in the model and prototype and the grain sizes can be directly scaled by the length scale, supporting Froude similarity.

St. Pierre Ostrander, T., Holzner, J., Mazzorana, B., Gorfer, M., Andreoli, A., Comiti, F., and Gems, B.: Confluence morphodynamics in Mountain Rivers in response to intense tributary bedload input, Earth Surf. Processes, 1-22, doi:10.1002/esp.5613, 2023.

Young, W. J., & Warburton, J. (1996). Principles and practice of hydraulic modelling of braided gravel-bed rivers. *Journal of Hydrology (NZ)*, *35*(2), 175–198.

(iii) Whether it was ensured that flow remains fully rough turbulent throughout

Response: This was an overlooked component of the manuscript, although considered during the model design and planning phase and certainly deserves mentioning, we appreciate the attentiveness of the reviewer. While turbulence was not evaluated during an experiment (the only real-time flow field measurement possible was free surface elevation), we assume that the flow is turbulent throughout the model domain, during an experiment. This is based on laminar flow generally occurring at lower flow velocities (Chow, 1959; Dey, 2014). In this regard most, open channel flows are turbulent (Chanson, 2004). A table (Table 1) has been included in the manuscript to show the hydraulic variables that would help indicate this where the velocities and associated Froude numbers suggest that the flow is, and remains turbulent throughout the model domain. These details were lacking in the initial submission and the addition should impart clarity. Furthermore, if we assume a simplified version of the physical experiment with a trapezoidal main channel, at the initial conditions (bed morphology is unchanged), without a tributary, the Reynolds number of the flow ranges from ~ 14000-116000, which shows the flow is fully rough turbulent, without the additional turbulence that is characteristic of confluences (Mosley, 1976; Best, 1987; Penna et al., 2018; De Serres et al., 1999). Flow turbulence has been added to the text L172.

Best, J. L.: Flow Dynamics at River Channel Confluences: Implications for Sediment Transport and Bed Morphology, in: Recent Developments in Fluvial Sedimentology, SEPM Special Publication, edited by: Etheridge, F.G., Flores, R.M. and Harvey, M.D., Society for Sedimentary Geology, Tulsa, OK, 39, 27-35, doi:10.2110/pec.87.39.0027, 1987

Chow, V.T.: Open Channel Hydraulics, McGraw-Hill, New York, ISBN 9780070859067, 1959.

Chanson, H. (2004). *The hydraulics of open channel flow*. Butterworth-Heinemann.

De Serres, B., Roy, A., Biron, P., and Best, J. L.: Three-dimensional flow structure at a river channel confluence with discordant beds, Geomorphology, 26, 313-335, doi:10.1016/S0169-555X(98)00064-6, 1999.

Dey, S. (2014). Turbulence in Open-Channel Flows. In: Fluvial Hydrodynamics. GeoPlanet: Earth and Planetary Sciences. Springer, Berlin, Heidelberg. doi: 10.1007/978-3-642-19062-9_3

Mosley, M. P.: An experimental study of channel confluences, J. Geol., 84(5), 535-562, doi:10.1086/628230, 1976.

Penna, N.; De Marchis, M.; Canelas, O.B.; Napoli, E.; Cardoso, A.H.; Gaudio, R.: Effect of the Junction Angle on Turbulent Flow at a Hydraulic Confluence. Water, 10, 469, doi:10.3390/w10040469, 2018.

(iv) More importantly, as D50 is about 2 mm, and D16 around 0.7 mm, there needs to be a discussion about scale effects on the threshold for entrainment (critical Shields number), given that it will not scale geometrically for smaller sediments. This effect might not be that important, because D84 may be large enough (6.0 mm) but still, this needs to be explained/discussed.

Response: The reviewer brings up an excellent point that certainly requires discussion. Physical models of all types are subject to some degree of scale effects (Chanson, 2004; Heller, 2011), the presented work being no different. The Shields ($\theta$) number and the grain Reynolds (Re*) number were calculated ($D_{50}$) in the main channel for all discharges and geometric configurations. At the lowest (15 l/s main and 1.5 l/s tributary) discharges $\theta$ and Re* at the model scale range from 0.08-0.10 and 60-67, respectively. At prototype scale $\theta$ and Re* range from 0.08-0.10 and 9849- 10927, respectively. At the next discharge combination (45 l/s and 4.5 l/s) $\theta$ and Re* at the model scale range 0.15-0.17 and 82-87. At prototype scale $\theta$ and Re* range from 0.15-0.17 and 13524-14247, respectively. While there is certainly a significant shift in Re* between lab and prototype scale, Aufleger (2006), for example, states that assuming Froude similarity and minimizing scale effects for pre-Alpine gravel bed rivers Re* numbers at the model scale above 80 are recommended. In this regard for the lowest discharge experiments the smaller grain fractions are subject to some degree of scale effects. We were aware of this when designing the model and scaling down the GSD curves obtained from confluences in nature. Scale limitations of grain size diameters are

discussed in Zarn (1992), where grain sizes smaller than 0.22 mm can change the flow-grain interaction due to cohesion effects. Oliveto and Hager (2005) discuss limiting the $D_{50}$ to 0.8 mm. The model grain size distribution has a minimum grain size of 0.5 mm, and a $D_{50}$ of 1.4 mm, adhering to published recommendations to minimize grain scale effects. According to Chanson (2004) if scale effects become significant a smaller prototype-to-model scale ratio should be considered. The physical model is designed to accommodate scale factors of 20-40 (a full description of the scaling method is discussed in St. Pierre Ostrander et al., 2023), this is not only a small prototype-to-model ratio but also is within scale model guidelines (Chanson, 2004) as recommended scale factors to deal (suggests 50:1 or 25:1) with the most likely scale effects, which are viscous scale effects. These combined factors show, that while the presented work is not free from some degree of scale effects, guidelines were followed and measures were taken to reduce them while still adhering to Froude similarity. Scale effects are discussed in detail in Section 4.2 of the discussion titled "Modelling limitations" (L584-610) as we fully agree with the reviewer that this a topic that requires a thorough discussion.

Aufleger, M. (2006): Flussmorphologische Modell. Grundlagen und Anwendungsgrenzen. Berichte des Lehrstuhls und der Versuchsanstalt für Wasserbau und Wasserwirtshaft, TU München. Band Nr. 104: 198-207. TU München.

Chanson, H. (2004). *The hydraulics of open channel flow*. Butterworth-Heinemann.

Oliveto, G., Hager, W.H.: Further results to time dependent local scour at bridge elements. J. Hydraulic Eng.131(2), 97–105, doi:10.1061/(ASCE)0733-9429(2005)131:2(97), 2005

St. Pierre Ostrander, T., Holzner, J., Mazzorana, B., Gorfer, M., Andreoli, A., Comiti, F., and Gems, B.: Confluence morphodynamics in Mountain Rivers in response to intense tributary bedload input, Earth Surf. Processes, 1-22, doi:10.1002/esp.5613, 2023.

Zarn, B.: Lokale Gerinneaufweitung: Eine Massnahmezur Sohlenstabilisierung der Emme bei Utzenstorf (Localriver expansion: A measure to stabilize the bed of Emme River at Utzendorf). VAW Mitteilung 118. D. Vischer ed.ETH Zurich, Zürich [in German], 1992.

A.2. In general, many of the figures, plots and tables are not at the adequate scale, or seem unnecessarily complicated, and/or do not have complete descriptions (captions).

(i) The "depositional geomorphic units" discussed in Section 3.1 deserve a specific figure, describing each one such units in more detail, at a better scale, so readers can understand exactly what is meant, visually. Figure should contain typical cross-sections and longitudinal profiles of the identified features or units.

Response: We agree that the specific geomorphic units should have more detail. Figure 3 has been revised with an emphasis on the geomorphic units, including longitudinal and transversal plots. Longitudinal plots start 1 m upstream of the confluence and end 6 m downstream of the confluence with profiles spaced every 0.1 m. Transversal profiles span 1m up and downstream of the confluence and focus on the confluence zone. The added plots help differentiate between geomorphic units, enhancing the clarity of the manuscript. Text has been added (L288-292 and L316-321) to describe the additional plots and how the geomorphic units were defined. Longitudinal and transversal plots for experiments 1-15 and a description of the resulting morphologies based on these plots is summarized in greater detail in St. Pierre Ostrander et al. (2023) including an appendix section with longitudinal plots of all depositional features.

St. Pierre Ostrander, T., Holzner, J., Mazzorana, B., Gorfer, M., Andreoli, A., Comiti, F., and Gems, B.: Confluence morphodynamics in Mountain Rivers in response to intense tributary bedload input, Earth Surf. Processes, 1-22, doi:10.1002/esp.5613, 2023.

(ii) Figures 3, 4, 7, A1 to A9 are all too small to allow for a clear visualization of what is happening at the confluence. Trying to show the total extent of both flumes makes it very difficult to clearly see what happens where it really matters. The focus should be on the confluence, showing the detail at a better scale (i.e., zoom into the confluence). This will probably make it impossible to keep all the figures in the paper, but I would rather see fewer of them, as long as they have the right combination of size and quality that allows me to visualize what is happening. Can't a "supplemental material file" include each one of the

figures you are now giving, at that smaller scale? I am sure there must be a better way; as it is now, even viewing the pdf file at 200% zoom does not allow me to resolve the details at the confluence in Figures 3 and 4, for example.

Response: Figure 3 has been revised (see response to (i)) to place more emphasis on the geomorphic units. Figure 4 and 7 have been revised to show only the lowest, middle, and maximum discharge experiments, therefore reducing the panel count to 8 and 9, respectively, allowing for larger images within the figure. Appendix 1-9 have been revised to increase the image size. For the appendix the full model domain is shown for all experiments because we want to show that there is a tributary response and to show the full morphological development as it progresses according to discharge. We feel that increasing the image size based on the reviewers' comments supports the number of panels in the appendix figures, while decreasing the number of panels in the figures in the body of the manuscript allows for a mix of a more focused assessment of key features while showing the global morphological response.

(iii) Related to the previous comment, on L.239-41 you state that "Deposition cones formed for all configurations and sediment concentrations when Q was 15 L/s and 1.5 L/s in the main channel and tributary, respectively." But this is not clear at all from the panels and images mentioned in L.232-33! Visually, it would seem that deposition cones were formed in all cases except A1 (d) & (e), A4 (c) to (e). Thus, I do not understand the statement.

Response: This aspect of the manuscript lacked clarity. The revised Figure 3 with longitudinal and transversal profiles better indicates the delineation of the geomorphic units. Deposition cones do not occupy the separation zone, have a compact longitudinal extent, steep gradients in both upstream and downstream directions, and the most deposition upstream of the confluence. This is apparent when examining the deposition peaks (y-axis maximum value) in Fig3d-Fig3f which show that with increasing discharge depositional patterns occur (from hydraulic forcing) further downstream. The transitional

morphology partially occupies the separation zone and, as shown in Fig 3e has a longitudinal profile that is a hybrid between the deposition cone and the fully developed bank attached bar. The bank-attached bar fully occupies the separation zone with a steep upstream gradient and a much less steep downstream gradient. Along with the revised Figure 3, text has been added L288-292 and L629-632 to clarify this aspect of the manuscript to better show the defining attributes of the geomorphic units.

(iv) In Table 3, both text and table header need better explanations of what is being given here. Please mention that 0.5% and 5% and 10% are the longitudinal slopes.

Table 3, now Table 4 has been completely revised in response to the comment L297.

(v) In Figs. 5, 6, 8, and 9, the combined use of two different types of charts (bars and scatterplots) to show the same type of differences is very confusing, visually.

Figure 5, 6 , 8, and 9 have been revised to better convey the results and add clarity.

**2. Minor comments:**

B.1. Why do the authors repeatedly refer to "sediment concentrations." Is there an expectation behind this language that only suspended sediment matters?

Response: The term sediment concentration (Hübl et al., 2018) deals with all sediment in the flow, although in this case sediment concentration is represented through bedload transport. Discussing sediment concentrations as it relates to coarse grains (not suspended) is fairly common for steep mountain channels (e.g., Arattano & Franzi, 2004; Church, 2015). By removing the finest fractions from the experimental GSD curve (see A1 iv) there was essentially no suspended sediment. It is not in terms of a dissolved or suspended sediment solution, but the concentration relative to the tributary discharge and is used to differentiate between flow types, movement types, processes, and behavior.

Arattano, M., & Franzi, L. (2004). Analysis of different water-sediment flow processes in a mountain torrent. *Natural Hazards and Earth System Sciences*, *4*(5/6), 783–791. doi:10.5194/nhess-4-783-2004

Church, M. (2015). Channel Stability: Morphodynamics and the Morphology of Rivers. In: Rowiński, P., Radecki-Pawlik, A. (eds) Rivers – Physical, Fluvial and Environmental Processes. GeoPlanet: Earth and Planetary Sciences. Springer, Cham. doi:10.1007/978-3-319-17719-9_12

Hübl, J. (2018): Conceptual Framework for sediment management in Torrents. Water, 10(12), 1718, doi:10.3390/w10121718.

B.2. The wordage "introduced factors" in 113, 118, 183, etc. is too general, and thus inadequate.

Response: "Introduced factors" has been changed to "introduced controlling factors" throughout the manuscript.

B.3. On L.214-15 you state "Volume samples were taken after an experiment with sample locations corresponding to both morphologic and hydraulic zones occurring in the channel." Shouldn't you state "after each experiment?" Also, please explain with more detail the reference to "both morphologic and hydraulic zones"

Response: "After an experiment" has been changed to "After each experiment" and references were added to provide additional details as to the typically occurring hydraulic and morphologic zones at river confluences L257-259.

B.4. On L.225-26, please indicate the values of Dmax and rho-s for your sediment

Response: The values for Dmax and rho-s have been added to lines 263-264.

B.5. At different locations in the ms, but mostly on L.231 and thereafter, you mention "unit stream power." I assume you refer to some kind of "length-wise averaged unit stream power," computed with the slope of the main flume and the sum of the discharges, right? And not to some local measure of $\Omega$, the unit

stream power at the location where each type of feature or unit occurs, computed with the local value of Sf, right?

More importantly, what does it mean to associate a single value of Omega to each type of unit? I ask because I do not understand the point of giving single values of average Ω corresponding to each experiment, in Table 3. What the table really does is not "associate units to unit stream power," but give the value of omega (as well as the geomorphic units) observed for each experiment. I also note that this is a "reach-scale averaged Ω" as discussed above, so what is the point of doing this? This should be explained better. You could actually add a column to Table 1 to give the values of Ω for each run...

Response: The concept of unit stream power has been used since the 1970s (Yang, 1971a, 1971b, 1971c) to explain the structure of natural stream networks, the formation and behavior of a meandering river, its longitudinal profile, and the formation of morphologic structure like rifles and pools. We used unit stream power ($\omega$) because sediment transport capacity along a stream is related to the energy dissipation per unit channel width. According to Bagnold (1960, 1966) unit stream power is the mean available power supply to the column of fluid over unit bed area whereas total stream power (Ω) is an expression for the rate of potential energy expenditure per unit length of channel (Knighton, 1999). Associating steam power values to specific morphologies (Bawa et al., 2014), patterns of sediment storage (Mcnab et al., 2006) and transport (Church, 2015), and classifying and predicting channel types and associated habitats (Flores et al., 2006) has been previously executed. Additionally, this practice has been applied at the catchment scale to predict channel adjustments; Bizzi and Lerner (2013) combined absolute and gradient values of total stream power and unit stream power to correctly predict the status of 87.5% of sites within their study catchment. Yochum et al. (2017) and Sholtes et al. (2018) found that the unit stream power was a significant predictor of the geomorphic response to the 2013 flooding in Colorado. By associating a single

value of unit stream power for a specific geomorphic unit a prediction tool is developed where, based on the initial conditions, the resulting morphology can be anticipated. The unit stream power for each channel was calculated based on the actual discharges for each run and with the combined measured discharges in the main channel. Calculations are based on the channel width at initial conditions (no deposition or aggradation). The combined discharge is also shown because the morphological development of the confluence largely occurs downstream of the confluence. The unit stream power values are therefore not averaged, but a unit stream power value at a specific cross-section, and because the channel at initial conditions is prismatic, the unit stream power at each cross section either upstream or downstream of the tributary is uniform and therefore subject to similar hydraulic conditions. It is a simplification to calculate unit stream power based on initial conditions but provides a method to link hydraulic conditions with depositional or erosional forms. However, we agree that this aspect of the manuscript requires clarity, accordingly, text has been added L275-L276, L297-300, and a revised table (Table 4) has been added which more clearly shows the experimental unit stream power values for both channels and the combined discharge section of the channel.

Bagnold, R. A., 1960, Sediment discharge and stream power--A preliminary announcement: US Geological Survey, Circular 421.

Bagnold, R. A., 1966, An approach to the sediment transport problem from general physics: US Geological Survey, Professional Paper 422

Bawa, N., Jain, V., Shekhar, S., Kumar, N., & Jyani, V. (2014). Controls on morphological variability and role of stream power distribution pattern, Yamuna River, western India. *Geomorphology*, *227*, 60–72. https://doi.org/10.1016/j.geomorph.2014.05.016

Bizzi, S., Tangi, M., Schmitt, R.J.P., Pitlick, J., Piégay, H. & Castelletti, A.F. (2021) Sediment transport at the network scale and its link to channel morphology in the braided Vjosa River system. Earth Surface Processes and Landforms, 46(14), 2946–2962, doi:/10.1002/esp.5225

Church, M. (2015). Channel Stability: Morphodynamics and the Morphology of Rivers. In: Rowiński, P., Radecki-Pawlik, A. (eds) Rivers – Physical, Fluvial and Environmental Processes. GeoPlanet: Earth and Planetary Sciences. Springer, Cham. doi:10.1007/978-3-319-17719-9_12

Flores, A. N., B. P. Bledsoe, C. O. Cuhaciyan, and E. E. Wohl (2006), Channel-reach morphology dependence on energy, scale, and hydroclimatic processes with implications for prediction using geospatial data, *Water Resour. Res.*, 42, W06412, doi:10.1029/2005WR004226.

Knighton, A. D. (1999). Downstream variation in stream power. *Geomorphology*, *29*(3–4), 293–306. https://doi.org/10.1016/s0169-555x(99)00015-x

Macnab, K., Jacobson, C., & Brierley, G. (2006). Spatial variability of controls on downstream patterns of sediment storage: A case study in the Lane Cove catchment, New South Wales, Australia. *Geographical Research*, *44*(3), 255–271. https://doi.org/10.1111/j.1745-5871.2006.00388.x

Sholtes, J.S., Yochum, S.E., Scott, J.A. & Bledsoe, B.P. (2018) Longitudinal variability of geomorphic response to floods. Earth Surface Processes and Landforms, 43(15), 3099–3113, doi:/10.1002/esp.4472.

Yang, C. T. 1971a. *Potential energy and stream morphology.* Water Resources Research, V. 7(2): 311-322.

Yang, C. T. 1971b. *On river meanders.* Journal of Hydrology, V. 13(3):231-253.

Yang, C. T. 1971c. *Formation of riffles and pools.* Water Resources Research, V. 7(6):1567-1574.

Yochum, S.E., Sholtes, J.S., Scott, J.A. & Bledsoe, B.P. (2017) Stream power framework for predicting geomorphic change: The 2013 Colorado Front Range flood. Geomorphology, 292, 178–192, doi:10.1016/j.geomorph.2017.03.004

B.6. What is exactly meant by "gradients of the deposited sediment" in L.306? This gradient is a local characteristic that will change continuously along and across the flume. This needs to be better explained.

Response: The gradients of deposited material in the channel refer to the depositional gradient after an experiment and were determined through a linear regression of the surface profile in the tributary channel. This method was also utilized in St. Pierre Ostrander et al. (2023) to determine the gradients of the depositional forms in the main channel as well as sediment deposited in the tributary channel, text has been added L362-363 to clarify this aspect.

St. Pierre Ostrander, T., Holzner, J., Mazzorana, B., Gorfer, M., Andreoli, A., Comiti, F., and Gems, B.: Confluence morphodynamics in Mountain Rivers in response to intense tributary bedload input, Earth Surf. Processes, 1-22, doi:10.1002/esp.5613, 2023.

B7. In the discussion on L.306-315 about bed aggradation/degradation, it seems to me that in the second case, that of aggradation in the tributary, the whole issue is a bit "artificial" and/or unrelated to the topic of study, as (i) it relates to the impact of procedural choices (feed rate), (ii) that happen outside the confluence. I am saying this because the aggradation in the tributary starts at its upstream end only because too much sediment is being fed there, which would not happen in an actual river system (unless there were another steep tributary contributing too much sediment to our "main tributary," or else the tributary had a sharp decrease in slope or increase in cross-section, just before its confluence with the main river).

Response: While the feed rate was the reason behind the aggradation at the upstream end of the tributary channel the feed rates are representative of fluviatile (Hübl et al., 2018) processes occurring in steep mountain channels. Torrential catchments are extremely variable and can host a range of processes based on sediment and flow regime interactions (Aulitzky, 1980; 1989; Meunier, 1991; Roca et al., 2009; Guillén-Ludeña et al., 2017; Hübl, 2018). Ideally, we would have had a much longer channel (a limitations section has been added to the discussion (4.2)) where sediment would be delivered to the tributary much further from the confluence to better represent natural conditions while avoiding any artificial depositional mechanisms, but space and funding were limiting factors. However, Church (2015) discusses a river achieving a regime condition. A regime condition is defined by stable river geometry which has undergone adjustments in response to the sediment load and hydraulic conditions. Church (2015) describes four possible adjustments:

1. adjustment of grain size of the sediments exposed on the bed surface, or of the dimensions of primary bedforms;

2. adjustment of channel width by bank erosion or sediment deposition in bars;

3. adjustment of channel macroform elements (the development of pools, riffles and bars);

4. adjustment of the mean gradient of the river by net degradation or aggradation.

In confined channels, which are typical of steep tributaries in developed areas (our study region), the only possible adjustments relate to grain size (1), and gradient (4) as macroform channel elements are not typical structures in steep, lined, supercritical channels and width adjustments are not possible. The response of the gradient of the deposited sediment is reflective of a flow with a saturated transport capacity. Transport efficiency on a fan requires steeper gradients when the hydrological regime is low where insufficient transport results in channel steepening due to sediment aggradation (Bowman, 2018) which was observed for experiments 16-30 with a 5% tributary gradient. Additionally, investigations into flooding occurring in steep tributaries have been carried out with adjusted sediment feed rates (Davies, 1990).

Bowman, D. (2018). Slope gradients. *Principles of Alluvial Fan Morphology*, 25–35. doi:10.1007/978-94-024-1558-2_3

Aulitzky, H.: Preliminary two-fold classification of debris torrents, Conference Proceedings of Interpraevent 1980, Bad Ischl, Austria, 8-12 September, (Vol. 4, 285-309, 1980, translated from German by G. Eisbacher) Internationale Forschungsgesellschaft, Interpraevent, Klagenfurt.

Aulitzky, H.: The debris flows of Austria, B. Eng. Geol. Environ., 40, 5-13, doi:10.1007/BF02590338, 1989.

Davies, T. R. H. (1990). Debris Flow Surges - Experimental Simulation. Journal of Hydrology (NZ), 29(1), 18–46.

Hübl, J. (2018): Conceptual Framework for sediment management in Torrents. Water, 10(12), 1718, doi:10.3390/w10121718.

Meunier, M.: Éléments d'hydraulique Torrentielle, Cemagref, France, EAN13 9782759212088, 1991.

Roca, M., Martín-Vide, J.P., and Martin-Moreta, P.: Modelling a torrential event in a river confluence, J. Hydrol., 364, 207-215, doi:10.1016/j.jhydrol.2008.10.020, 2009.

B8. Because you have the previous discussion about deposition/erosion (aggradation/degradation) in the tributary, as discussed in Comment B.7, maybe it would be a good idea to remind the reader at the beginning of Section 3.4, in the Overview (Subsection 3.4.1), that all response variables here refer to the main channel, not the tributary.

Response: This distinction was overlooked but has been added to L421.

B9. Discussion on lines 433-36: More than "the degree of turbulence increasing," I would think that this is caused by the increased lateral or transversal momentum with respect to the main, longitudinal direction (which in turn should result in increased turbulence). I don't understand the following comment about "improved mixing." IMO, it is the other way around: when the confluence angle is decreased, the flows align more closely (the transversal momentum is decreased) which should result in less mixing between the two flows, not more.

Response: Convergence and mutual flow deflection of confluent flows can generate secondary currents (Lewis et al., 2020). The lateral momentum from the tributary channel influences the large-scale helical flow motion produced by the curvature of the stream lines (Mosley, 1976; Lewis et al., 2020). Helical motion generates vertical velocity components that resemble flow behavior in a meander bend. The levels of turbulence in the junction are largely associated with the shear layer formed in the main channel between tributary and main channel flows (Guillén Ludeña et al., 2017). The increased turbulence arises from the increased mutual flow deflection where powerful vertical vortices are generated along the shear layer, which increase the bed shear stress within the junction, resulting in considerable bed scour (Guillén Ludeña et al., 2017; Best, 1987). Reducing the confluence angle decreases the turbulence generated along the shear layer allowing for improved mixing of flows (across the shear layer) in the confluence producing shallower scour. Mixing of flows has been experimentally evaluated with dye injections where the tributary and main channel flow remain generally separate in the confluence zone but became completely

mixed downstream of the confluence (Mosley, 1976). We state that reducing the confluence angle improves mixing, which decreases turbulence in the junction, which is an established effect of changing the confluence angle (Best, 1987) and is related to the flow segregation caused by mutual flow deflection. As the confluence angle increases the two streams undergo a progressively greater deflection to flow parallel to the post-confluence channel (Best, 1987). Furthermore, Penna et al. (2018) investigate the effects of the confluence angle specifically for mountain river conditions. They conclude that with a higher junction angle, the stagnation zone at the upstream junction corner and the separation zone become wider and longer, increasing flow deflection at the entrance of the tributary into the post-confluence channel. Mixing is less when the confluence angle is greater because the two flows must mix across the shear layer between the confluent streams, which at higher confluence angles and discharge ratios is more intense. Text has been added to clarify this component L69-82, L493-496 and the revised discussion section which also discusses this aspect has been revised (Section 4) L511-532.

Best, J. L.: Flow Dynamics at River Channel Confluences: Implications for Sediment Transport and Bed Morphology, in: Recent Developments in Fluvial Sedimentology, SEPM Special Publication, edited by: Etheridge, F.G., Flores, R.M. and Harvey, M.D., Society for Sedimentary Geology, Tulsa, OK, 39, 27-35, doi:10.2110/pec.87.39.0027, 1987

Guillén Ludeña, S., Z. Cheng, G. Constantinescu, and M. J. Franca (2017), Hydrodynamics of mountain river confluences and its relationship to sediment transport, J. Geophys. Res. Earth Surf., 122, 901–924, doi:10.1002/2016JF004122

Lewis, Q., Rhoads, B., Sukhodolov, A., & Constantinescu, G. (2020). Advective lateral transport of streamwise momentum governs mixing at small river confluences. *Water Resources Research*, 56, e2019WR026817. doi:10.1029/2019WR026817

Mosley, M. P.: An experimental study of channel confluences, J. Geol., 84(5), 535-562, doi:10.1086/628230, 1976.

Penna, N.; De Marchis, M.; Canelas, O.B.; Napoli, E.; Cardoso, A.H.; Gaudio, R.: Effect of the Junction Angle on Turbulent Flow at a Hydraulic Confluence. Water, 10, 469, doi:10.3390/w10040469, 2018.

B10. "When more channel was available," on L.457. Can you explain this better, please?

Response: It refers to the separation zone not occupying as much of the channel when the confluence angle was 45° relative to the size of the separation zone when the confluence angle was 90°. Text has been added to clarify this statement L531-532.

B11. Sentence on L.515-17: (i) Writing issue in that you can't say "Tending towards sth was characterized…;" you need a noun (e.g., "The tendency towards" or "The evolution .."). (ii) More importantly, the meaning is not clear when you state that this "Tendency towards … was characterized by the geomorphic units… ." Please re-write giving more details, indicating what exactly varies with the geomorphic units (I mean: you stated before that you obtained the same units across all experiments so now that you say that these units "are indicators of the flood magnitude" – which varies – you must indicate what is specifically changing with the units). Also, maybe "reflected" is a better word than "characterized?"

Response: All writing suggestions have been implemented, which has improved the quality of the conclusion. The conclusion has been re-written with details that describe what exactly is happening. Specifically, with increasing discharge, the geomorphic units transitioned from a cone to a bank-attached bar as the depositional patterns were forced further downstream and into the separation zone, with the bank-attached bar occupying the full extent of the separation zone L614-L638.

B12. Your single sentence about future work, L.523-24, does not contain sufficient information to understand how "ecologically valuable measures" such as "sediment buffer zones" relate to the work at hand (beyond the fact that they are related to sediment).

Response: The conclusion has been re-written to focus more on the presented work and to avoid any confusion regarding potential outlooks and future work L616-L640.

**3. Writing suggestions/changes:**

L.111: laser scans?

An overlooked typo that has been revised L133.

L.128: … but considers an additional case for the tributary gradient as well as for the confluence angle

This helpful writing suggestion has been applied to the manuscript L155-157

L.132: …while the tributary channel had a fixed bed

The writing suggestion has been applied L161.

L.157:  … a more natural river bed, while the post-run ….

The writing suggestion has been applied L197.

L.177:  Experiment 30 could not be completed…

The writing suggestion has been applied L220.

L.186:  "subtle variations" in what, exactly?

"Morphological" has been added to the text L229.

L.222: grain size analysis

"Size" has been added to the text L268.

L.288-89: … of morphological differences at confluences caused by different tributary gradients"

The figure caption has been re-written L339-L341.

L.361: "Statistical evidence of…" or else "Statistical analysis of …?"

Statistical analysis has been added to the section title L418.

L.375: Please replace "provoked" by "resulted in" or "caused" or sth else

Provoked has been replaced with resulted in L433.

L.379-80: Please rephrase this sentence, as the three main ideas (clauses) are not well linked logically

The sentence has been re-written L436-438.

L.380: concentration instead of concentration; same thing on L.519

The typo has been changed L439 and L634

L.382: semi-colon instead of comma

The punctuation has been revised L441.

L.411-12: tributary-transported sediment

The punctuation suggestion has been applied L470

L.416: Add semi-colon to "… on the response variables; (sigma) is the standard …"

A semi colon has been added L474.

L.417: Add semi-colon immediately after D

A semi colon has been added L475.

L.438: "two" instead of 2? And a semi-colon before sigma

All suggestions have been added to the text L498.

L.441: Delete comma

Comma deleted L502.

L.443 and 445: The reference to Fig. 9 seems wrong

Thanks for catching this reference error, the figure should be 8 and has been revised L503 L505.

L.451: Maybe add some nuance to the sentence, like "… as one of the main drivers of confluence morphology, thus affecting the spatial distribution of hydraulic zones, in the case of lowland confluences"?

This helpful writing suggestion has been applied to the manuscript L511-512.

L.462-63: Please flip the sentence to get the grammar right. Maybe "It has been previously observed in mountain rivers (citations) that the tributary channel gradient responds to …"?

The writing suggestion has been implemented in the manuscript L536-538.

L.474-75: Maybe "Adjustments to sediment concentration were reflected in varying ranges for deposition and erosion depths and volumes, as well as varying extents for these geomorphic units." ("these" refers to the mention in the previous sentence that the same units occur across experiments)

The manuscript has been revised utilizing this suggestion L553-554

L.484: Replace "the transport capacity" by "its transport capacity?"

The suggestion has been added to the text L563

L.508: I would just say "… the findings of the literature dealing with …"

The suggestion has been implemented in the re-written conclusion L616.

L.514-15: Replace the second "as" with a "while?" Can't have "… as …. as …"

The sentence has been revised L627

L.559: Should say "the geomorphic units" (plural); also, "than" instead of "then"

Both suggestions have implements L674-675.

Many locations: I don't know whether this is an EGU thing, but I see that you separate numbers from %, e.g., "5 % tributary gradient," but at the same time keep numbers together to units, e.g., "above or below 0.01m." The rules I am used to are the exact opposite: "the channel has a 5% slope and is 12 m wide."

We have adjusted spacing to follow the suggestion of the reviewer.